# Localization of Na+ channel clusters in narrowed perinexi of gap junctions enhances cardiac impulse transmission via ephaptic coupling: a model study

Ena Ivanovic [ID] and Jan P. Kucera [ID]

*Department of Physiology, University of Bern, Bern, Switzerland*

Edited by: Bjorn Knollmann & Eleonora Grandi

Linked articles: This article is highlighted in a Perspectives article by Veeraraghavan *et al*. To read this article, visit https://doi.org/10.1113/JP282350.

The peer review history is available in the Supporting Information section of this article (https://doi.org/10.1113/JP282105#support-information-section).

**Abstract**   It has been proposed that when gap junctional coupling is reduced in cardiac tissue, action potential propagation can be supported via ephaptic coupling, a mechanism mediated by negative electric potentials occurring in narrow intercellular clefts of intercalated discs (IDs). Recent studies showed that sodium (Na+) channels form clusters near gap junction plaques in nanodomains called perinexi, where the ID cleft is even narrower. To examine the electrophysiological relevance of Na+ channel clusters being located in perinexi, we developed a 3D finite element model of two

**Ena Ivanovic** is currently a PhD candidate in the Integrative Cardiac Bioelectricity Group at the Department of Physiology of the University of Bern led by Professor Jan P. Kucera. She has a background in Mechanical Engineering with a double Master's degree from the RWTH Aachen University (Germany) and the Ecole Centrale de Nantes (France). Ena is keen to apply her knowledge to life sciences and her research focuses on the development of new modelling approaches to address open research questions on impulse propagation in cardiac tissue.

The Journal of Physiology

longitudinally abutting cardiomyocytes, with a central Na$^+$ channel cluster on the ID membranes. When this cluster was located in the perinexus of a closely positioned gap junction plaque, varying perinexal width greatly modulated impulse transmission from one cell to the other, with narrow perinexi potentiating ephaptic coupling. This modulation occurred via the interplay of Na$^+$ currents, extracellular potentials in the cleft and patterns of current flow within the cleft. In contrast, when the Na$^+$ channel cluster was located remotely from the gap junction plaque, this modulation by perinexus width largely disappeared. Interestingly, the Na$^+$ current in the ID membrane of the pre-junctional cell switched from inward to outward during excitation, thus contributing ions to the activating channels on the post-junctional ID membrane. In conclusion, these results indicate that the localization of Na$^+$ channel clusters in the perinexi of gap junction plaques is crucial for ephaptic coupling, which is furthermore greatly modulated by perinexal width. These findings are relevant for a comprehensive understanding of cardiac excitation.

(Received 1 July 2021; accepted after revision 6 September 2021; first published online 17 September 2021)

**Corresponding author** Jan P. Kucera: Department of Physiology, University of Bern, Bühlplatz 5, CH-3012 Bern, Switzerland. Email: jan.kucera@unibe.ch

**Abstract figure legend** To examine the relevance of sodium channel clusters being located in perinexi (regions surrounding gap junction plaques in cardiac intercalated discs), we developed a finite element model (mesh shown on top) of two longitudinally abutting cardiomyocytes, with a central sodium channel cluster on the intercalated disc membranes. In the example shown, when the sodium channel cluster (red) was located in the perinexus (grey) of a closely positioned non-permeable gap junction plaque (green), impulse transmission occurred via ephaptic coupling (bottom left). In contrast, when the sodium channel cluster was located remotely from the gap junction plaque and thus outside the perinexus, no transmission occurred (bottom right). Thus, perinexi are privileged sites for ephaptic coupling.

## Key points

- Ephaptic coupling is a cardiac conduction mechanism involving nanoscale-level interactions between the sodium (Na$^+$) current and the extracellular potential in narrow intercalated disc clefts.
- When gap junctional coupling is reduced, ephaptic coupling acts in conjunction with the classical cardiac conduction mechanism based on gap junctional current flow.
- In intercalated discs, Na$^+$ channels form clusters that are preferentially located in the periphery of gap junction plaques, in nanodomains known as perinexi, but the electrophysiological role of these perinexi has never been examined.
- In our new 3D finite element model of two cardiac cells abutting each other with their intercalated discs, a Na$^+$ channel cluster located inside a narrowed perinexus facilitated impulse transmission via ephaptic coupling.
- Our simulations demonstrate the role of narrowed perinexi as privileged sites for ephaptic coupling in pathological situations when gap junctional coupling is decreased.

## Introduction

Every contraction of the heart is coordinated by action potentials (APs) that propagate through electrically communicating myocytes. Classically, the mechanism of cardiac impulse propagation relies on gap junctional proteins (connexins) forming low-resistance connections between cardiomyocytes, whereby rapidly activating voltage gated sodium (Na$^+$) channels depolarizing the myocytes at the wave front provide the driving force for the excitation of downstream cells (Weidmann, 1970; Kleber & Rudy, 2004; Rohr, 2004). Many experimental and modelling studies (for a review, see Kleber & Rudy, 2004) have established the relationship between conduction velocity and intercellular resistance. However, some experimental findings are at odds with the classical theory of AP propagation. For instance, it was observed that connexin 43 knockout mice exhibit slowed ventricular conduction, but at a velocity that is still much higher than what would be anticipated based on the almost complete absence of gap junctions (Gutstein *et al.* 2001). One possible explanation for this discrepancy is given by the hypothesis that a second complementary conduction

mechanism, today known as ephaptic coupling, is involved in cardiac conduction when gap junctional coupling is reduced (Sperelakis & Mann, 1977; Sperelakis, 2002; Mori *et al.* 2008; Veeraraghavan *et al.* 2014a).

Ephaptic coupling involves the extracellular nanodomain between adjacent cells and relies on the presence of $Na^+$ channels in intercalated discs (IDs). When a cell is excited and $Na^+$ channels on the ID membrane are activated, $Na^+$ ions pass from the narrow extracellular cleft into this cell. According to Ohm's law, this relatively large $Na^+$ current ($I_{Na}$) flowing through the large resistance of the narrow cleft causes a substantial negative extracellular potential ($V_e$) in the cleft. In turn, this negative $V_e$ accelerates the activation of $Na^+$ channels and also activates $Na^+$ channels in the ID membrane of the neighbouring cell by rendering its transmembrane potential ($V_m$) less negative. Thus, this self-activating process can maintain cardiac conduction (Sperelakis & Mann, 1977; Veeraraghavan *et al.* 2014b).

This mechanism was corroborated by modelling studies from different research groups (Kucera *et al.* 2002; Mori *et al.* 2008; Hand & Peskin, 2010; Lin & Keener, 2010; Tsumoto *et al.* 2011; Veeraraghavan *et al.* 2015, 2016; Wei *et al.* 2016). These studies also indicated that another mechanism occurs, known as self-attenuation, whereby the negative $V_e$ also brings $V_m$ closer to the Nernst potential of $Na^+$, which decreases $I_{Na}$ and slows conduction. Optical mapping studies in isolated hearts, in which extracellular ion concentrations were changed and gap junction uncouplers were used, have yielded results that agree with the predictions of models incorporating ephaptic interactions (Veeraraghavan *et al.* 2015, 2016; King *et al.* 2021). Using patch clamp experiments, we recently showed that restricting the extracellular space near cells expressing $Na^+$ channels affects $I_{Na}$ in a manner compatible with self-activation and self-attenuation (Hichri *et al.* 2018).

At the morphological level, several studies have shown that $Na^+$ channels are preferentially expressed in IDs (Cohen, 1996; Maier *et al.* 2002). Immunohistochemical studies revealed that about 50% of the $Na^+$ channels are located in the IDs, where they form clusters (Lin *et al.* 2011; Shy *et al.* 2013; Leo-Macias *et al.* 2016). In parallel, it was demonstrated that these $Na^+$ channel clusters are found in the vicinity of gap junction plaques, where the ID cleft is typically narrower (Rhett & Gourdie, 2012; Rhett *et al.* 2013; Veeraraghavan *et al.* 2014b; Veeraraghavan & Gourdie, 2016; Raisch *et al.* 2018). The close proximity of $Na^+$ channel clusters and gap junction plaques led to the concept of 'perinexus', a nanoscale domain where channels aggregate and possibly interact (Rhett *et al.* 2013; Veeraraghavan *et al.* 2015, 2018).

However, the physiological relevance of $Na^+$ channel clusters being located in perinexi has, to our knowledge, never been thoroughly examined. In a previous study, we investigated the electrophysiological consequences of $Na^+$ channel clustering and demonstrated in a 2D finite element model of the ID that the size and position of $Na^+$ channel clusters have a major influence on ephaptic coupling (Hichri *et al.* 2018). However, this 2D model did not permit to assess the additional role of nearby gap junction plaque localization.

Therefore, in the present work, we developed a high-resolution 3D finite element model of a pair of longitudinally abutting cardiac cells. Our new model permits the incorporation of structural details at the nanoscale level and allows investigating both intra- and extracellular currents and potentials. We investigated the electrophysiological importance of a $Na^+$ channel cluster being localized close to a gap junction plaque on AP transmission via ephaptic coupling and how ephaptic coupling is affected by the width of the cleft in the bulk of the ID and in the perinexus. In the absence of perinexal cleft narrowing, the co-localization of the $Na^+$ channel cluster and the gap junction plaque affected ephaptic coupling only minimally and only for particular cleft widths. In contrast, the presence of a narrowed perinexus greatly potentiated ephaptic effects when the $Na^+$ channel cluster was positioned near the gap junction plaque. Our study thus reveals an important physiological function of the perinexus.

## Methods

### General principles

Our computational model is based on the finite element method, which is the method of choice to discretize (i.e. mesh) and simulate structures having arbitrary geometries. This represents a crucial advantage over the finite difference method, which, although being easier to implement, is limited to simple rectangular shapes and meshes (Sundnes *et al.* 2006). To model a cell pair joined by an ID, we developed two paradigms, called the explicit cleft model and the collapsed cleft model. In the explicit cleft model (Fig. 1*A*), the intra- and extracellular spaces (including the ID cleft) are represented explicitly by 3D domains separated by 2D cell membranes and appropriate boundary conditions are applied (Fig. 1*B*). The collapsed cleft model is at the same time an extension and a simplification of the explicit cleft model. In the collapsed cleft model (Fig. 1*C*), the 3D ID cleft and the two corresponding membranes are collapsed onto a 2D surface, on which three potentials coexist: the intracellular potentials of the two adjoined cells and the extracellular potential in the ID cleft. Collapsing the cleft, which is justified by its narrowness (orders of magnitude smaller than cell size), leads to important advantages (e.g. greatly reduced number of finite elements and hence accelerated computations) and represents the

main technical innovation in our approach compared to previous work (Agudelo-Toro & Neef, 2013; Horgmo Jæger *et al*. 2021).

Both paradigms are based on the physical law of charge conservation. Because the biophysical and mathematical description of the collapsed cleft model builds upon the explicit cleft model, both approaches are presented sequentially below. Briefly, the complete problem is described by a set of partial differential equations with appropriate boundary conditions, which is then reformulated as a set of coupled ordinary differential equations and solved using appropriate numerical methods, which were developed for this purpose. In preliminary simulations, the explicit cleft model was used to

validate the collapsed cleft model. All simulations shown in this work were conducted using the collapsed cleft model. Transmembrane ion currents were incorporated according to the well-known Hodgkin-Huxley formalism (Hodgkin & Huxley, 1952). Importantly, our model fully represents the intracellular potentials, the extracellular potentials (in the cleft and in the bulk space) and the transmembrane potentials. Thus, it provides a comprehensive picture of the bioelectrical phenomena occurring at a subcellular scale.

The detailed methods are presented below for interested readers. A more general reader can skip the rest of this section. Main model parameters and variables are listed in Tables 1 and 2, respectively.

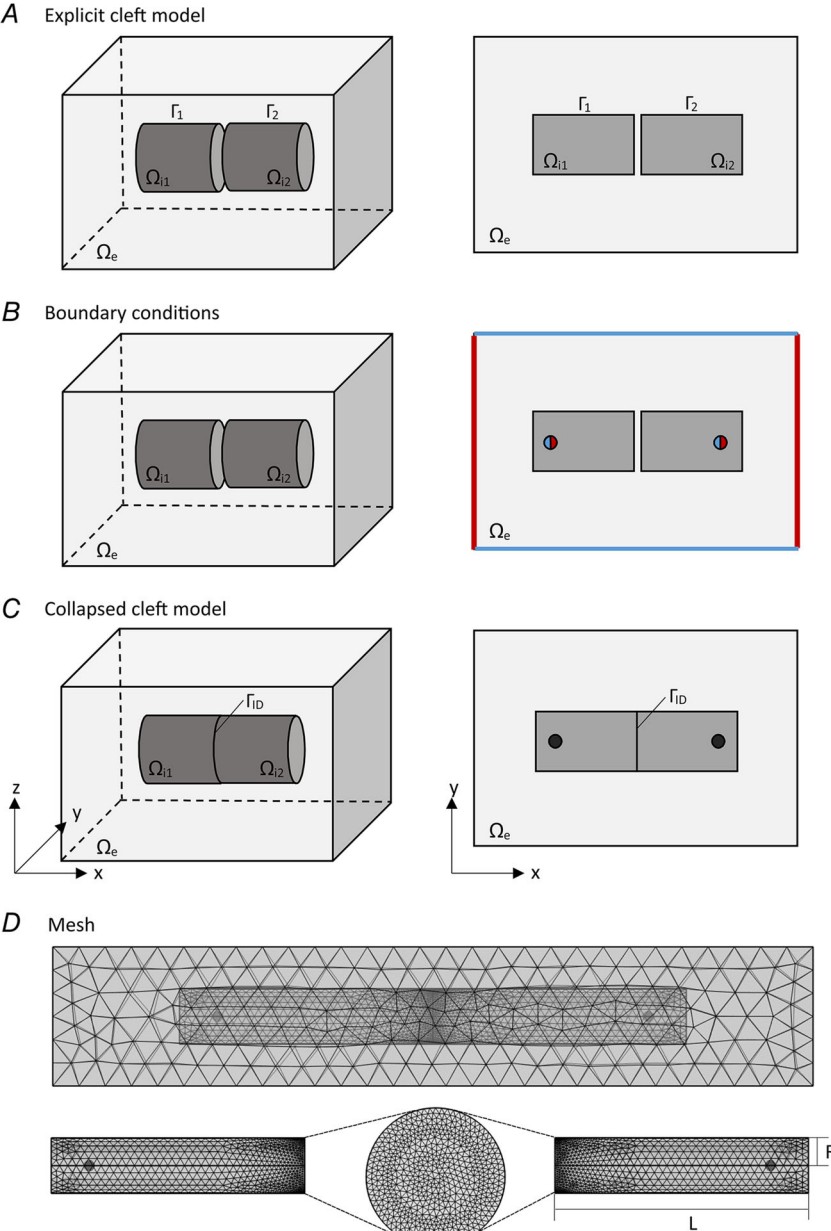

**Figure 1. Typical geometrical framework in three dimensions (left) with a 2D section in the *x–y* plane (right)**
The model consists of two intracellular domains labelled $\Omega_{i,k}$ ($k \in \{1,2\}$), corresponding to two cells. In the default model, two cylindrical cells of equal size are juxtaposed along the *x*-axis and the cleft between them is centred at *x* = 0. The intracellular space is surrounded by a rectangular cuboid, representing the extracellular space $\Omega_e$. Intra- and extracellular spaces are separated by the membrane $\Gamma_k$. *A*, explicit cleft model (note that the cleft width is greatly exaggerated). The cleft width is varied by changing the relative position of both cylinders to each other. *B*, Dirichlet (red) and/or Neumann (blue) boundary conditions applied as explained in the text. The boundaries of the extracellular domain perpendicular to the *x*-axis are grounded (Dirichlet condition with $V_e = 0$). On the circular/spherical patches inside the intracellular space a specific voltage or current clamp protocol is applied (corresponding to a Dirichlet or a Neumann condition, respectively). *C*, collapsed cleft model, in which the pre- and post-junctional membranes as well as the cleft are collapsed onto a 2D surface $\Gamma_{ID}$. *D*, top: mesh of the entire collapsed cleft framework. Bottom: meshes of the two cylindrical cells (cylinder length *L*, radius $R_{ID}$) interconnected by the 2D mesh on the surface $\Gamma_{ID}$ (with a uniform finite element density in this example).

**Table 1. List of the main parameters of the model including name, definition, value and unit**

| Name | Definition | Value | Unit |
|---|---|---|---|
| $d$ | Distance between the Na$^+$ channel cluster and the gap junction plaque | varied | $\mu$m |
| $R_{ID}$ | Radius of the cell and ID | 11 | $\mu$m |
| $r_{GJ}$ | Radius of the gap junction plaque | 0.6875 | $\mu$m |
| $r_{Na}$ | Radius of the Na$^+$ channel cluster | 0.6875 | $\mu$m |
| $r_{peri}$ | Radius of the perinexus | 2.26875 | $\mu$m |
| $L$ | Length of the cell | 100 | $\mu$m |
| $L_x$ | Length of the extracellular box | 300 | $\mu$m |
| $L_y$ | Width of the extracellular box | 55 | $\mu$m |
| $L_z$ | Height of the extracellular box | 55 | $\mu$m |
| $\sigma_e$ | Extracellular conductivity | 6.666 | mS/cm |
| $\sigma_{cleft}$ | Extracellular cleft conductivity | 6.666 | mS/cm |
| $\sigma_i$ | Intracellular conductivity | 3.333 | mS/cm |
| $\sigma_{gap}$ | Gap junctional conductivity | varied | mS/cm |
| $G_{gap}$ | Overall gap junctional conductance | varied | mS |
| | nominal 100% value | 0.00253 | mS |
| $C_m$ | Membrane capacitance | 1 | $\mu$F/cm$^2$ |
| $\bar{g}_{Na}$ | Maximal conductance for Na$^+$ | 23 | mS/cm$^2$ |
| $\bar{g}_K$ | Maximal conductance for K$^+$ | 0.3 | mS/cm$^2$ |
| $E_{Na}$ | Nernst potential for Na$^+$ | 55 | mV |
| $E_K$ | Nernst potential for K$^+$ | $-85$ | mV |
| $w$ | Cleft width | varied | nm |
| $f_\sigma$ | Factor for extracellular conductivity | 1 | unitless |
| $F_{gNa,ID}$ | Scaling factor for Na$^+$ channel density on the ID membranes | 5.05 | unitless |
| $F_{gNa,lat}$ | Scaling factor for Na$^+$ channel density on the lateral membranes | 0.555 | unitless |
| $\alpha$ | Parameter for the numerical method | 0.5 | unitless |
| $\beta$ | Parameter for the numerical method | 0.5 | unitless |

**Table 2. List of the main variables of the model including name, definition and unit**

| Name | Definition | Unit |
|---|---|---|
| $x$ | Position | cm |
| $t$ | Time | ms |
| $V_{e,i}$ | Extracellular / intracellular potential | mV |
| $V_m$ | Membrane potential | mV |
| $I_m$ | Membrane current density | $\mu$A/cm$^2$ |
| $I_{ion}$ | Transmembrane ionic current density | $\mu$A/cm$^2$ |
| $I_{gap}$ | Gap junctional current density | $\mu$A/cm$^2$ |
| $m$ | Activation gating variable of the Na$^+$ current | unitless |
| $h$ | Fast inactivation gating variable of the Na$^+$ current | unitless |
| $j$ | Slow inactivation gating variable of the Na$^+$ current | unitless |

## Model description

As a starting point, we use the computational method proposed by Agudelo-Toro & Neef (2013), which allows the simulation of intracellular, extracellular and membrane potentials in realistic cell geometries. As shown in Fig. 1*A*, the model space consists of extracellular ($\Omega_e$) and intracellular ($\Omega_i$) domains being segregated from each other ($\Omega = \Omega_e \cup \Omega_i$). Both domains do not overlap ($\Omega_e \cap \Omega_i = 0$), but join at the membrane $\Gamma$. The intracellular domain $\Omega_i$ consists of two cells (represented schematically as cylinders in Fig. 1*A*) with their ID membranes facing each other and forming explicitly a narrow cleft. The explicit cleft is part of the extracellular

domain $\Omega_e$, which is modelled as a rectangular box around the intracellular domains $\Omega_i$.

A quasi-static approximation of Maxwell equations leads to the following expressions, reflecting charge conservation:

$$-\nabla(\sigma_e(x)\nabla V_e(x,t)) = \rho_e \text{ in } \Omega_e, \tag{1}$$

$$-\nabla(\sigma_i(x)\nabla V_i(x,t)) = \rho_i \text{ in } \Omega_i, \tag{2}$$

with $x$ being the position ($x \in \mathbb{R}^3$), $V_i$ and $V_e$ being the intra- or extracellular potential, respectively and $\rho_{i,e}$ being current volume density source terms (that can be functions of space and time; in our work, we considered them to be zero unless specified otherwise). Intra- and extracellular domains are characterized by space-dependent conductivity tensors $\sigma_i$ and $\sigma_e$, respectively. In the present work, we assume that both domains are isotropic. We set $\sigma_e$ to a scalar value of 6.666 mS/cm, which is equivalent to a resistivity of 150 $\Omega$cm (Shaw & Rudy, 1997). Assuming that the intracellular space is roughly half as conductive as the extracellular space (due to the presence of the contractile proteins of the cardiomyocytes), we set $\sigma_i$ to 3.333 mS/cm corresponding to a resistivity of 300 $\Omega$cm.

Intra- and extracellular domains are separated from each other by a membrane $\Gamma$, which is assumed to be of negligible (zero) thickness. Membrane potential $V_m$ is defined as the potential difference at the same point between the intra- and extracellular domains:

$$V_m = V_i(x) - V_e(x) \text{ on } \Gamma. \tag{3}$$

On the membrane $\Gamma$, the normal component of the extra- and intracellular current density is continuous

and defined as the total membrane current density $I_m$:

$$n_e\,\sigma_e\,\nabla V_e = -n_i\,\sigma_i\,\nabla V_i = I_m \text{ on } \Gamma, \qquad (4)$$

with $n_e$ being the normal vector pointing out from $\Omega_e$ and $n_i$ the normal vector pointing out from $\Omega_i$. The total membrane current density $I_m$ corresponds to the sum of the capacitive current density and the transmembrane ionic current density ($I_{ion}$):

$$I_m = C_m\,dV_m/dt + I_{ion}, \qquad (5)$$

where $C_m = 1\mu\text{F}/\text{cm}^2$ is the membrane capacitance per unit area.

The main model parameters and variables are presented in Tables 1 and 2, respectively.

## Spatial discretization

Space is discretized with the finite element method. In order to obtain the weak or variational formulation of the problem, the partial differential equations (PDEs) are multiplied with test functions $\varphi_{i,e}$ (corresponding to the extra- and intracellular potential) and $u$ (corresponding to membrane potential), belonging to an appropriate function space. The PDEs are integrated over the domain $\Omega$ and Green's identity is applied, giving:

$$\int_{\Omega_e} \sigma_e \nabla V_e \nabla \varphi_e \, dV - \int_{\Gamma} I_m \varphi_e \, dA = \int_{\Omega_e} \rho_e \varphi_e \, dV, \qquad (6)$$

$$\int_{\Omega_i} \sigma_i \nabla V_i \nabla \varphi_i \, dV + \int_{\Gamma} I_m \varphi_i \, dA = \int_{\Omega_i} \rho_i \varphi_i \, dV, \qquad (7)$$

$$\int_{\Gamma} V_m\, u \, dA = \int_{\Gamma} (V_i - V_e)\, u \, dA, \qquad (8)$$

where $dA$ refers to a surface and $dV$ to a volume integral. Equation (8) is the weak formulation of membrane potential $V_m$, expressed as a boundary condition (eqn 3). The sum of eqns (6) and (7) forms the complete system of spatial equations:

$$\int_{\Omega} \sigma \, \nabla V \, \nabla \varphi \, dV + \int_{\Gamma} I_m [\varphi]_{i,e} \, dA = \int_{\Omega} \rho \, \varphi \, dV, \qquad (9)$$

where the operator $[]_{i,e}$ is positive for the integration performed on the intracellular and negative on the extracellular side of the membrane $\Gamma$.

In order to formulate the original PDEs in terms of a system of ordinary differential equations in matrix form, a set of piecewise linear basis functions is chosen and the continuous functions are replaced by discrete

formulations of the problem (with $i$, $j$, $k$ and $l$ being indices), leading to:

$$\sigma \sum_k V^k \int_{\Omega} \nabla\varphi^k \, \nabla\varphi^i \, dV + \sum_j I_m^j \int_{\Gamma} u^j \left[\varphi^i\right]_{i,e} \, dA = \int_{\Omega} \rho \, \varphi^i \, dV, \quad (10)$$

$$\sum_k V^k \int_{\Gamma} u^j \left[\varphi^k\right]_{i,e} \, dA = \sum_l V_m^l \int_{\Gamma} u^l \, u^j \, dA. \qquad (11)$$

By defining $K^{ik} = \sigma \int_{\Omega} \nabla\varphi^k \nabla\varphi^i \, dV$, $B^{ij} = \int_{\Gamma} u^j [\varphi^i]_{i,e} \, dA$, $f^i = \int_{\Omega} \rho \, \varphi^i \, dV$ and $G^{jl} = \int_{\Gamma} u^l u^j \, dA$, we obtain the following expressions:

$$\sum_k V^k K^{ik} + \sum_j I_m^j B^{ij} = f^i, \qquad (12)$$

$$\sum_k V^k B^{kj} = \sum_l V_m^l G^{jl}. \qquad (13)$$

This system of eqns (12) and (13) can then be written in the matrix form $M\,u = b$, as done by Agudelo-Toro & Neef (2013) as well as by Ying & Henriquez (2007):

$$\begin{bmatrix} K & B \\ B^T & 0 \end{bmatrix} \begin{bmatrix} V \\ I_m \end{bmatrix} = \begin{bmatrix} f \\ GV_m \end{bmatrix}. \qquad (14)$$

The $2 \times 2$ block matrix $M$ in eqn (14) consists of a so-called stiffness matrix $K$ and the matrices $B$ and $B^T$.

By a suitable ordering of the finite element vertices (nodes), the vector of potentials $V$ can be put in the block form $V = [V_e \; V_i \; V_{em} \; V_{im}]^T$, where $V_e$ and $V_i$ are extra- and intracellular potentials at points that are not on $\Gamma$ and $V_{em}$ and $V_{im}$ are extra- and intracellular potentials at corresponding membrane points, respectively. Based on the similarity of the definitions of $B$ and $G$, the matrix $B$ can then be expressed in the following block structure, with $G$ and $-G$ reflecting the change of sign due to the operator $[]_{i,e}$:

$$B = \begin{bmatrix} 0 \\ 0 \\ -G \\ G \end{bmatrix} \qquad (15)$$

This step changes the matrix formulation into:

$$\begin{bmatrix} & & & 0 \\ & K & & 0 \\ & & & -G \\ & & & G \\ 0 & 0 & -G & G & 0 \end{bmatrix} \begin{bmatrix} V_e \\ V_i \\ V_{em} \\ V_{im} \\ I_m \end{bmatrix} = \begin{bmatrix} f_e \\ f_i \\ f_{em} \\ f_{im} \\ GV_m \end{bmatrix}, \qquad (16)$$

where $f = [f_e\ f_i\ f_{em}\ f_{im}]^T$ consists of corresponding source terms. These terms are considered zero for a voltage clamp protocol, but can be considered non-zero at nodes corresponding to Neumann boundary conditions for current clamp protocols.

In the following, we show how to simplify the matrix formulation in eqns (14) and (16) based on the structure of the matrix $B$ (eqn 15). These simplifications are performed in order to obtain a sparser symmetric matrix $M$, which will ultimately allow the implementation of a numerical integration method based on sparse symmetric positive definite matrices. The system of equations (eqn 14) is formed by:

$$KV + BI_m = f, \tag{17}$$

$$B^T V = GV_m. \tag{18}$$

The matrix $G$ (corresponding to a mass matrix) is by construction positive definite and therefore invertible. In a first step, the matrix $G$ is inverted and eqn (18) is left-multiplied by $G^{-1}$, giving:

$$G^{-1}B^T V = V_m. \tag{19}$$

Defining $U = B(G^{-1})^T = [0\ 0\ -I\ I]^T$ and $U^T = G^{-1}B^T = [0\ 0\ -I\ I]$, eqn (18) can then be written as follows:

$$U^T V = V_m. \tag{20}$$

This last equation states that at a given point on the membrane, $V_m = V_i - V_e$, equivalently to eqn (3).

Considering the last matrix row in eqn (16), we obtain eqn (21), which is then left-multiplied by $G^{-1}$, giving eqn (22):

$$-GV_{em} + GV_{im} = GV_m, \tag{21}$$

$$-V_{em} + V_{im} = V_m. \tag{22}$$

Furthermore, considering the second and third last rows of the matrix formulation in eqn (16), we introduce the substitution $Z = G I_m$. The matrix formulation is then transformed to:

$$\begin{bmatrix} & & & 0 \\ & & & 0 \\ & K & & -I \\ & & & I \\ 0 & 0 & -I & I & 0 \end{bmatrix} \begin{bmatrix} V_e \\ V_i \\ V_{em} \\ V_{im} \\ Z \end{bmatrix} = \begin{bmatrix} f_e \\ f_i \\ f_{em} \\ f_{im} \\ V_m \end{bmatrix}. \tag{23}$$

In this way, with the use of positive and negative identity matrices, the block matrix $M$ can be formed with the matrices $U$ and $U^T$ instead of $B$ and $B^T$. These steps lead to the following system of equations, with a matrix $M$ being simpler and sparser (note that explicit inversion of $G$ is not required):

$$\begin{bmatrix} K & U \\ U^T & 0 \end{bmatrix} \begin{bmatrix} V \\ GI_m \end{bmatrix} = \begin{bmatrix} f \\ V_m \end{bmatrix}, \tag{24}$$

with : $U = \begin{bmatrix} 0 \\ 0 \\ -I \\ I \end{bmatrix}$ and $U^T = \begin{bmatrix} 0 & 0 & -I & I \end{bmatrix}$. $\tag{25}$

Introducing the formulation for the membrane current (eqn 5), the first row of eqn (24) and thus the matrix formulation becomes:

$$K V + U G C_m\, dV_m/dt = f - U G I_{ion}, \tag{26}$$

$$\begin{bmatrix} K & U \\ U^T & 0 \end{bmatrix} \begin{bmatrix} V \\ Q\, dV_m/dt \end{bmatrix} = \begin{bmatrix} f - U G I_{ion} \\ V_m \end{bmatrix}, \tag{27}$$

with $Q = \sum_j C_m \int_\Gamma u^j \varphi^i ds$. $\tag{28}$

Of note, if $C_m$ is uniform, $Q = C_m G$.

## Incorporation of boundary conditions

Dirichlet (D) and/or Neumann (N) boundary conditions need to be applied on the exterior boundary $d\Omega_e$ as well as on the inside boundary of the intracellular domain $d\Omega_i$:

$$V_e = V_D (x, t) \text{ on } d\Omega_D, \tag{29}$$

$$\sigma_e \nabla V_e\, n_e = I_N \text{ on } d\Omega_N. \tag{30}$$

On two sides of the extracellular box Dirichlet conditions of zero potential (ground) are applied (see Fig. 1*B*). Cellular electrophysiological experiments are usually conducted with micropipettes under voltage clamp or current clamp conditions. Rather than simulating pipettes and their geometries explicitly, we simulate the opening of micropipettes as spherical regions within the intracellular domains of the cells, as illustrated in Fig. 1*B*. This corresponds to cellular impalements, in which the pipette tips are inserted into the cytoplasm. A voltage clamp protocol is then implemented as a Dirichlet condition on the corresponding spherical surface. Conversely, a current clamp protocol is implemented as a uniform current source on the corresponding surface. Thus, whether Dirichlet of Neumann boundary conditions are used for the spherical patches of the intracellular domain depends on the applied patch clamp protocol (voltage or current clamp). For a current clamp protocol, the Neumann boundary source term $I_N$ will be set to a non-zero value.

Once the boundary conditions are defined, the matrix $M$ is rearranged in the following $3 \times 3$ block form by adequate permutation of rows and columns:

$$K = \begin{bmatrix} K_D & K_x \\ K_x^T & K_N \end{bmatrix}; \quad U = \begin{bmatrix} U_D \\ U_N \end{bmatrix}, \quad (31)$$

$$\begin{bmatrix} K_D & K_x & U_D \\ K_x^T & K_N & U_N \\ U_D^T & U_N^T & 0 \end{bmatrix} \begin{bmatrix} V_D \\ V_N \\ Q \, dV_m/dt \end{bmatrix} = \begin{bmatrix} I_D \\ f_N - U_N \, G \, I_{ion} \\ V_m \end{bmatrix}, \quad (32)$$

where the subscripts D and N refer to Dirichlet and non-Dirichlet nodes, respectively and the subscript X labels the off-diagonal blocks of $K$.

Considering in eqns (33) and (34) the two last rows of the matrix formulation (eqn 32) and rearranging eqn (33), a $2 \times 2$ block matrix system (eqn 35) can be extracted from the $3 \times 3$ block matrix system:

$$K_x^T V_D + K_N V_N + U_N Q \frac{dV_m}{dt} = f_N - U_N \, G \, I_{ion}, \quad (33)$$

$$U_D^T V_D + U_N^T V_N = V_m, \quad (34)$$

$$\begin{bmatrix} K_N & U_N \\ U_N^T & 0 \end{bmatrix} \begin{bmatrix} V_N \\ Q \, dV_m/dt \end{bmatrix} = \begin{bmatrix} f_N - K_x^T V_D - U_N \, G \, I_{ion} \\ V_m \end{bmatrix}. \quad (35)$$

This system of ordinary differential equations must now be solved numerically for the potentials at the non-Dirichlet nodes ($V_N$) and the membrane potentials ($V_m$).

## Time discretization

Here, we implement time discretization based on a flexible rule proposed by Sundnes et al. (2006), which depends on the values of two parameters $\alpha$ ($0 \leq \alpha \leq 1$) and $\beta$ ($0 \leq \beta \leq 1$) satisfying $\alpha + \beta = 1$.

First, membrane potential $V_m$ is expressed as a weighted average of its value at the present time $t$ and at the next time step $t + \Delta t$, giving eqn (36). Then, the time steps $t$ and $t + \Delta t$ are separated from each other, leading to eqn (37).

$$V_m = U_N^T V_N = \alpha V_m(t) + \beta V_m(t + \Delta t), \quad (36)$$

$$V_m(t + \Delta t) = \frac{1}{\beta} U_N^T V_N - \frac{\alpha}{\beta} V_m(t). \quad (37)$$

The time derivative of membrane potential $V_m$ is considered using a finite difference approximation:

$$\frac{dV_m}{dt} = \frac{V_m(t + \Delta t) - V_m(t)}{\Delta t} \quad (38)$$

Different choices of the values of $\alpha$ and $\beta$ lead to differences in accuracy and characteristics of the numerical method. For all choices of $\alpha$ and $\beta$, the relation $\alpha + \beta = 1$ has to be valid. The explicit forward Euler method is obtained for $\alpha = 1$ and thus $\beta = 0$. For a setup with $\alpha = 0$ and $\beta = 1$, the scheme is recognized as the implicit backward Euler method. Optimal stability properties and accuracy are achieved by setting $\alpha = \beta = 0.5$, resulting in the Crank-Nicolson (CN) method (Crank & Nicolson, 1974). Compared to all other combinations of $\alpha$ and $\beta$, the CN method is known for its best accuracy, allowing longer time steps while remaining stable (Sundnes et al. 2006). Furthermore, a shortening of computational time of can be achieved (Sundnes et al. 2006). These advantageous characteristics led us to use the semi-implicit time iteration scheme based on the CN method for the further development of the numerical integration method. The application of the CN method on the matrix formulation (eqn 35) is as follows. The first row of eqn (35) is:

$$K_N V_N + U_N Q \frac{dV_m}{dt} = f_N - K_x^T V_D - U_N \, G \, I_{ion}, \quad (39)$$

The combination of eqns (38) and (39) lead to eqn (40):

$$K_N V_N + \frac{1}{\Delta t} U_N Q V_m(t + \Delta t) - \frac{1}{\Delta t} U_N Q V_m(t)$$
$$= f_N - K_x^T V_D - U_N \, G \, I_{ion}. \quad (40)$$

Furthermore, the expression for the membrane potential at time step $t + \Delta t$ (eqn 37) is used to derive eqn (41):

$$K_N V_N + \frac{1}{\Delta t} U_N Q \left( \frac{1}{\beta} U_N^T V_N - \frac{\alpha}{\beta} V_m(t) \right)$$
$$= f_N - K_x^T V_D - U_N \, G \, I_{ion} + \frac{1}{\Delta t} U_N Q V_m(t). \quad (41)$$

Rearranging eqn (41) gives the final expression for the numerical method based on the CN scheme (eqn 42):

$$\left( K_N + \frac{1}{\Delta t \, \beta} U_N Q U_N^T \right) V_N = \frac{1}{\Delta t} \left( 1 + \frac{\alpha}{\beta} \right) U_N Q V_m(t)$$
$$+ f_N - K_x^T V_D - U_N \, G \, I_{ion}. \quad (42)$$

The solution at the time step $t$ is known, whereas the solution at the time step $t + \Delta t$ is sought. During every time step the already known solution for $V_m$ is used to compute

the right hand side of eqn (42). Then, the linear system 42 is solved to obtain the next value of $V_N$ and the next value of $V_m$ is obtained from $V_m = U^T V_N$. Of great importance is the fact that the matrix on the left hand side of eqn (42) is symmetric and positive definite, which allows using specific algorithms such as Cholesky factorization or the conjugate gradient method (Sundnes *et al.* 2006). In the present work, we use Cholesky factorization. Since the matrix does not change with time (for a given constant time step $\Delta t$), the Cholesky factorization needs to be computed only once at the beginning of a simulation.

## Collapsing the explicit cleft

The explicit representation of the physiological ID cleft between two cardiac cells (see Fig. 1*A*) leads to several challenges when the geometry is divided into finite elements. Firstly, due to the narrowness of the cleft (around 20 nm; Raisch *et al.* 2018) in comparison to the size of a cell being in the micrometre range, a high density of very small finite elements is required in order to mesh the extracellular space between two cells as well as the intracellular space in the vicinity of the ID. As a result, poor element quality and singularity effects at corners may occur. Thus, the explicit 3D cleft model requires a large number of nodes in the region of interest leading to a large computational effort (time and memory). Secondly, a new problem arises when one attempts to model gap junction plaques in the ID. Gap junction plaques are composed of densely packed intercellular channels consisting each of two hemichannels (connexons) contributed by each respective cell (Kleber & Rudy, 2004). At such plaques (having a size in the range of 100–1000 $\mu$m) the intercellular space is only 2–4 nm wide (Pitts, 1978) and the remaining extracellular space between the connexons is drastically reduced, but probably not inexistent. To incorporate gap junctions into the explicit cleft model would then require the definition of new conductive domains belonging to both the intracellular and extracellular spaces. Moreover, the very narrow cleft width at gap junctions would further augment the necessary number and density of nodes.

To circumvent these problems and with the aim of allowing the use of larger elements with less nodes, we propose an alternative modelling approach in which the cleft is reduced to an entity of lower dimension. The explicit ID cleft, represented by a volume between the two cells, is reduced to a folded (and possibly tortuous) 2D structure (manifold) $\Gamma_{ID}$ embedded in three dimensions. On this 2D domain, three values co-exist: the intercellular potential $V_i$ in the first and second cell ($V_{i1}$ and $V_{i2}$) and the extracellular potential $V_e$ in the cleft. We call this the collapsed cleft model, which is illustrated in Fig. 1*C* and *D*.

In addition to eqns (1)–(5), the collapsed cleft model is described further by the following equations valid on $\Gamma_{ID}$:

$$-\nabla_{\Gamma_{ID}} \left( w \, f_\sigma \, \sigma_e \, \nabla_{\Gamma_{ID}} V_e \right)$$

$$= I_{ion,1} + C_m \frac{dV_{m,1}}{dt} + I_{ion,2} + C_m \frac{dV_{m,2}}{dt}, \quad (43)$$

$$-n_{i,1} \, \sigma_{i,1} \, \nabla V_{i,1} = I_{m,1} + I_{gap \, 1\rightarrow 2}, \quad (44)$$

$$-n_{i,2} \, \sigma_{i,2} \, \nabla V_{i,2} = I_{m,2} + I_{gap \, 2\rightarrow 1}, \quad (45)$$

$$I_{gap \, a\rightarrow b} = \sigma_{gap} \, (V_a - V_b), \quad (46)$$

where $w$ is the cleft width, $f_\sigma$ is a factor permitting to adjust extracellular conductivity relative to that of bulk extracellular space and $\sigma_{gap}$ is the gap junctional conductance (per unit surface). The ion current density $I_{ion,k}$ the transmembrane potential $V_{m,k}$ and the normal vector $n_{i,k}$ pointing outward of the intracellular space are defined for the first and second cell using the indices k $\in$ {1, 2}, respectively.

The first equation (eqn 43) is similar to the one that we proposed previously for a 2D ID (Hichri *et al.* 2018), with the difference that the operator $\nabla_{\Gamma_{ID}}$ must be understood as the 2D gradient in the tangent space of $\Gamma_{ID}$. In this formulation, $w$, $\sigma_{gap}$ and $f_\sigma$ can be functions of position in $\Gamma_{ID}$, which permits modelling of heterogeneous cleft widths, specific distributions of gap junction plaques and different conductivities in gap junction plaques as well as plicate and interplicate regions of the ID. In addition, $V_e$ must be continuous at the junction $J$ between $\Omega_e$ and $\Gamma_{ID}$ ($J$ is a 1D curve embedded in three dimensions). Equations (44) and (45) now include the current flow through the gap junctions, with the gap junctional current density $I_{gap \, a\rightarrow b}$ depending on the direction of flow and being defined by eqn (46).

For the collapsed cleft model, the overall mesh consists of four separate sub-meshes including the first and the second cell, the extracellular bulk without the cleft and the ID cleft itself (Fig. 1*D*). For a practical application using finite elements, the first step is to concatenate the arrays of extracellular nodes and to eliminate the duplicate nodes belonging to $J$, which will ensure continuity of $V_e$ on $J$. Due to the difference in the dimensionality of $\Omega_e$ and $\Gamma_{ID}$, the corresponding elements (e.g. tetrahedra in $\Omega_e$ and triangles in $\Gamma_{ID}$) need to be treated separately to construct the stiffness matrix $K$. The non-cleft contribution $K_{nc}$ (of the bulk extracellular space and the intracellular spaces) to $K$ is calculated as in eqn (12). The contribution $K_c$ of the cleft space $\Gamma_{ID}$ is calculated in a similar way (considering the conductivity in the cleft $\sigma_{cleft}$ multiplied by the cleft width $w$), but by using integration over the 2D manifold $\Gamma_{ID}$ (the collapsed cleft) rather over the 3D space. A similar integration over the 2D manifold $\Gamma_{ID}$ and with

respect to the gap junctional conductivity $\sigma_{\text{gap}}$ is used to calculate $M_{\text{gap}}$, the mass matrix that corresponds to $\sigma_{\text{gap}}$ in eqn (46). In order to define the contribution of the gap junction plaques $K_{\text{gap}}$ to $K$, $M_{\text{gap}}$ is pasted on the elements of the sparse matrix $K_{\text{gap}}$ corresponding to a connection between matching intracellular membrane points in cells 1 and 2 (a connection bridging the ID from one intracellular space to the other). For the diagonal of $K_{\text{gap}}$, we use $M_{\text{gap}}$ (by definition positive definite) to ensure $K$ is positive definite, whereas $-M_{\text{gap}}$ is placed on the non-diagonal nodes, leading to a sparse matrix $K_{\text{gap}}$:

$$K_{\text{gap}} = \begin{bmatrix} M_{\text{gap}} & \cdots & -M_{\text{gap}} \\ \vdots & \ddots & \vdots \\ -M_{\text{gap}} & \cdots & M_{\text{gap}} \end{bmatrix}. \qquad (47)$$

Then, $K$ is represented as the sum of three components:

$$K = K_{\text{nc}} + K_{\text{c}} + K_{\text{gap}}. \qquad (48)$$

Thus, the stiffness matrix $K$ is the sum of three matrices $K_{\text{nc}}$, $K_{\text{c}}$ and $K_{\text{gap}}$. $K_{\text{nc}}$ is obtained from all the 3D (e.g. tetrahedral) elements of the intra- and extracellular domains. $K_{\text{c}}$ and $K_{\text{gap}}$ are obtained from all the 2D (e.g. triangular) elements of the domain $\Gamma_{\text{ID}}$. $K_{\text{nc}}$ determines current flow in the bulk extracellular space and the intracellular domains, $K_{\text{c}}$ determines current flow within the ID cleft and $K_{\text{gap}}$ determines current flow from one side of $\Gamma_{\text{ID}}$ to the other.

If the intracellular nodes (of both cells) contiguous to $\Gamma_{\text{ID}}$ and the nodes of $\Gamma_{\text{ID}}$ itself match (i.e. coincide), a system of ordinary differential equations similar to eqns (14)–(28) can then be constructed and solved using the same numerical approach as described in eqns (31)–(42). This matching of mesh nodes can be guaranteed by suitable configuration of a meshing algorithm or software.

## Incorporation of ion currents

As we are interested principally in studying the dynamics of the Na$^+$ current and its interaction with intra- and extracellular potentials during cellular excitation (the AP upstroke), we use, as in our previous work (Hichri *et al.* 2018), a simplified electrophysiological model consisting only of the Na$^+$ current and a linear potassium (K$^+$) current (ensuring resting membrane polarization). In our model the total ion current density ($I_{\text{ion}}$) is included as the sum of Na$^+$ ($I_{\text{Na}}$) and K$^+$ current ($I_{\text{K}}$) densities:

$$I_{\text{ion}} = I_{\text{Na}} + I_K. \qquad (49)$$

Further ion current densities, such as the L-type calcium current, can be considered in the cell membrane

model by adding them in eqn (49). Each ion current ($I_{\text{Na}}$ and $I_{\text{K}}$) is expressed using Ohm's law:

$$I_{\text{X}} = g_{\text{X}} \, (V_{\text{m}} - E_{\text{X}}) \text{ for X} \in \{\text{Na, K}\}, \qquad (50)$$

with $E_{\text{x}}$ being the Nernst potential of passive ion transport (eqn 51) and $g_{\text{x}}$ being the permeability (conductance) of the membrane to the given ion species (eqns 52 and 53). The Nernst potentials are:

$$E_{\text{X}} = \frac{RT}{zF} \ln \frac{[\text{X}]_{\text{e}}}{[\text{X}]_{\text{i}}}, \qquad (51)$$

where $R$ is the ideal gas constant, $T$ the absolute temperature, $F$ the Faraday constant, $z$ the valence of the ion species and $[\text{X}]_{\text{e,i}}$ the respective extra- and intracellular ion concentration (Plonsey & Barr, 2007). In our model, the Nernst potentials were set to $E_{\text{Na}} = +55$ mV and $E_{\text{K}} = -85$ mV.

In order to describe the behaviour of $I_{\text{Na}}$, we used a Hodgkin & Huxley (1952) formalism according to Luo & Rudy (1991) with modifications by Livshitz & Rudy (2009). This formalism introduces three gating variables (probabilities): $m$, representing activation gates (there are three gates in the model), $h$ being the fast inactivation gate and $j$ being the slow inactivation gate (according to Beeler & Reuter, 1977). These gating variables define the dynamical behaviour of Na$^+$ channels. The three gating variables were integrated using the method of Rush & Larsen (1978). This leads to the formulation for Na$^+$ conductance shown in eqn (52). The K$^+$ current $I_{\text{K}}$ was modelled in a simplified manner (without gating) with the conductance $g_{\text{K}}$ as a constant (eqn 53).

$$g_{\text{Na}} = \bar{g}_{\text{Na}} \, m^3 \, h \, j, \qquad (52)$$

$$g_K = \bar{g}_K. \qquad (53)$$

The parameters $\bar{g}_{\text{Na}}$ (23 mS/cm$^2$) and $\bar{g}_K$ (0.3 mS/cm$^2$) represent the maximal conductance (per unit area) of Na$^+$ and K$^+$ channels, respectively.

Because Na$^+$ channels are non-uniformly expressed on the cell membrane (higher density in IDs than on the lateral membranes; Cohen, 1996; Kucera *et al.* 2002), $\bar{g}_{\text{Na}}$ needs to be multiplied with scaling factors given by the following equations:

$$F_{\text{gNa,ID}} = P \, \frac{A_{\text{cell}}}{A_{\text{ID}}}, \qquad (54)$$

$$F_{\text{gNa,lat}} = (1 - P) \frac{A_{\text{cell}}}{A_{\text{lat}}}. \qquad (55)$$

These scaling factors $F_{\text{gNa,ID}}$ and $F_{\text{gNa,lat}}$ are determined using the ratio of the area of the whole cell $A_{\text{cell}}$ and the area of the membranes adjacent to intercalated discs $A_{\text{ID}}$ (eqn 54) or the membrane on the lateral side (eqn 55),

respectively. For the calculation of the area, we used typical values for the length ($L = 100 \ \mu$m) and the radius ($R_{ID} = 11 \ \mu$m) of a cardiac cell (Lin *et al.* 2011). The proportion $P$ of channels located in the ID was set to 0.5 according to recent immunohistochemical studies, revealing that about 50% of the Na$^+$ channels are located in the ID (Lin *et al.* 2011; Shy *et al.* 2013; Veeraraghavan *et al.* 2015; Leo-Macias *et al.* 2016). Under these assumptions, the scaling factors for the ID and lateral membranes were $F_{gNa,ID} = 5.05$ and $F_{gNa,lat} = 0.555$, respectively. With the use of these scaling factors, the proportional distribution of Na$^+$ channels on the cell membrane is varied, but the total number of channels is kept constant. A similar approach was used to model gap junction plaques, as detailed in the Results section.

### Practical implementation

All simulations presented in this article were conducted using the collapsed cleft model. However, we used the explicit cleft model to validate the collapsed cleft model during its development. The 3D models (with explicit or collapsed cleft) were implemented using a hybrid approach combining the advantages of MATLAB (version 2019a, The MathWorks, Natick, MA, USA) and COMSOL Multiphysics (version 5.5, COMSOL AB, Stockholm, Sweden). COMSOL was used to generate geometries and corresponding meshes (in this work, we used unstructured tetrahedral and triangular meshes). The description of the meshes, including the coordinates of the nodes (vertices) and the element connectivity, were

then exported from COMSOL as text files and imported into MATLAB, where all the necessary matrices were computed based on the physiological parameters and boundary conditions and stored in sparse format. To process the meshes, we used some MATLAB functions from the package 'DistMesh' by Persson & Strang (2004). Simulations of potentials and ion currents were run in MATLAB based on the numerical methods presented above. As stated above, gating variables were integrated using the method of Rush & Larsen (1978). A time step $\Delta t$ of 0.0001 ms was used in all simulations.

## Results

### Na$^+$ channel cluster and gap junction plaque co-localization in the ID affect ephaptic interactions for very specific cleft widths

In a previous computational study, we demonstrated that Na$^+$ channel clustering in the IDs is a key component for signal transmission via ephaptic coupling (Hichri *et al.* 2018). When Na$^+$ channel clusters on the ID membranes faced each other across the cleft, ephaptic interactions were greatly enhanced and permitted AP transmission even for low (or zero) levels of gap junctional coupling. However, in that study, we did not evaluate the physiological consequences of the known fact that Na$^+$ channel clusters assemble in the proximity of gap junction plaques (Rhett & Gourdie, 2012; Rhett *et al.* 2013; Veeraraghavan & Gourdie, 2016). To investigate the additional role of co-localized gap junction plaques we formulated two hypotheses, which are illustrated in Fig. 2. In our first

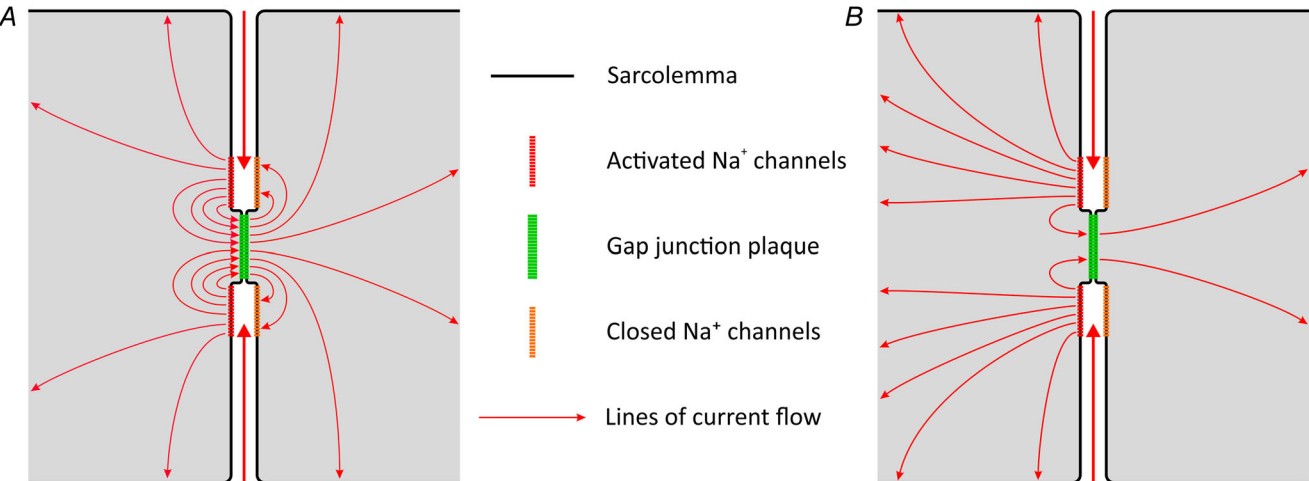

**Figure 2. Hypotheses regarding current fluxes through Na$^+$ channel clusters and an adjacent gap junction during ephaptic coupling**
*A*, the current from the extracellular cleft entering the first cell (left) through open Na$^+$ channels (red) flows mostly through the adjacent gap junction (green) into the second cell (right), where it activates the closed Na$^+$ channels (orange). Only a small part of the current distributes over the bulk membranes. *B*, in contrast, most of the current flowing into the first cell serves to depolarize its bulk membrane, whereas only a small part passes through the gap junction, where it is also redistributed to the membrane.

hypothesis (Fig. 2*A*), we postulated that a substantial part of the large $Na^+$ current entering the pre-junctional cell is channelled through the adjacent gap junction into the post-junctional cell, where it directly activates nearby $Na^+$ channels, which would greatly potentiate ephaptic coupling. In our alternate hypothesis (Fig. 2*B*), only a small part of the inward current passes through the gap junction, whereas most of the current redistributes and serves to depolarize the bulk membranes of both cells, thereby bearing less influence on ephaptic coupling. These hypotheses differ by the patterns of intracellular current flow and associated intracellular potential gradients; in the first hypothesis, larger gradients would be expected in the vicinity of the $Na^+$ channel clusters and the gap junction. To ascertain these phenomena, we used our collapsed cleft model of a cell pair, in which the intracellular spaces are represented explicitly.

For the ID membranes of the cardiac cells, we designed three different configurations (Fig. 3*A*), where one $Na^+$ channel cluster was consistently placed in the centre of the ID membrane of the first and the second cells. The $Na^+$ channel cluster radius was set to $r_{Na} = 0.6875$ $\mu$m which is equivalent to 1/16 of the ID radius $R_{ID}$; this is consistent with physiological sizes of such clusters ($r_{Na} < 1$ $\mu$m; Leo-Macias *et al.* 2016). All $Na^+$ channels of the IDs were placed inside these clusters, ensuring a constant total number of $Na^+$ channels (i.e. total cellular maximal $Na^+$ current density) for both cells. The first ID configuration corresponds to a reference configuration, where intercellular coupling (connexins) was distributed homogeneously in the ID around the centred $Na^+$ channel cluster. For the two other ID configurations, intercellular coupling was concentrated in a disc-shaped plaque (gap junction) of the same size as the $Na^+$ channel cluster ($r_{GJ} = r_{Na} = 0.6875$ $\mu$m). As illustrated in Fig. 3*A*, the gap junction plaque was either placed in close proximity (distance between centres: $d = 2r_{GJ}$) or at a larger distance ($d = 5r_{GJ}$) to the $Na^+$ channel cluster. For the second and third configurations, in the gap junction plaque, the cleft width ($w_{cleft}$) was set to 2 nm and the extracellular conductivity was reduced to 0.666 mS/cm (10% of nominal $\sigma_e$). This reduced $\sigma_e$ accounts for the dense packing of connexons in gap junctions (Flores *et al.* 2012), which is likely to decrease the overall conductance of the extracellular space. The remainder of the ID cleft was assigned a predefined width ranging from 10 to 100 nm. For the first configuration (no clustering of gap junctional coupling), cleft width was not narrowed to 2 nm and the nominal $\sigma_e$ was used. All three ID configurations were evaluated for cleft widths between 10 nm and 100 nm and for reduced levels of gap junctional coupling. The overall gap junctional conductance ($G_{gap}$) was set to 0%, 1%, or 5% of the normal 100% value (calculated to be 2.53 $\mu$S assuming an overall junctional resistivity of 1.5 $\Omega cm^2$ (Shaw & Rudy, 1997)). In the

second and third configurations, this conductance was then concentrated into the gap junction plaques. Of note, considering that there are many plaques between two coupled cardiomyocytes, the reduced conductances of 25.3 nS and 126.5 nS (1% and 5% of normal, respectively) are probably similar to the conductance of a single functional junctional plaque in the ID.

For all simulations, a Neumann boundary condition corresponding to a current clamp protocol with a rectangular current pulse (intensity: 11.5 nA, duration: 0.5 ms) was applied on the spherical patch of the pre-junctional cell (cell 1), where it elicited an AP. A Neumann condition was also applied on the spherical patch of the post-junctional cell (cell 2), mimicking a current clamp protocol with zero current. The resulting intracellular potentials ($V_i$) on the patches, the $Na^+$ currents ($I_{Na}$) on the ID and bulk membranes and the minimal intercellular potential (min $V_e$) in the cleft were recorded as a function of time. Note that the same finite element mesh was used for all three configurations (Fig. 3*A*) to exclude the possibility that different behaviours may be an artefact of using different meshes.

Figure 3*B* illustrates the results for zero gap junctional coupling, i.e. non-permeable gap junctions. In the context of the hypotheses being tested (Fig. 2), these simulations serve as controls. Of note, the properties of the extracellular cleft were nevertheless altered by the presence of the gap junction plaques, even if these plaques were not connecting the intracellular domains. Thus, in the absence of gap junctional coupling, the observed differences between the three configurations are due to differences in the conductivities of the extracellular space within the cleft. For cleft widths of 20, 40 and 70 nm, three contrasting responses of cell 2 resulted from the current pulse injected in cell 1. Irrespective of the $Na^+$ channel cluster–gap junction plaque arrangement, a very narrow cleft of 20 nm only led to a subthreshold depolarization of cell 2. Although the $Na^+$ current was activated in the ID membrane of cell 2, this current did not suffice to bring the bulk membrane of cell 2 to threshold. Thus, as a whole, cell 2 was not excited. In contrast, suprathreshold depolarization occurred for a 40 nm wide cleft, followed by an AP upstroke in cell 2. Because gap junctional conductivity $G_{gap}$ was set to zero, ephaptic coupling was the only mechanism responsible for AP transmission. For wider intercellular clefts (e.g. 70 nm), no change in intracellular potential occurred in the second cell. Interestingly, we observed nearly identical results for the three ID configurations when cleft width was set to 20 or 70 nm. Only for a 40 nm wide cleft were the results with varying ID configurations distinguishable. Indeed, the AP upstroke of the cell 2 was more delayed when the gap junction plaque and the $Na^+$ channel cluster were located next to each other compared to the two other

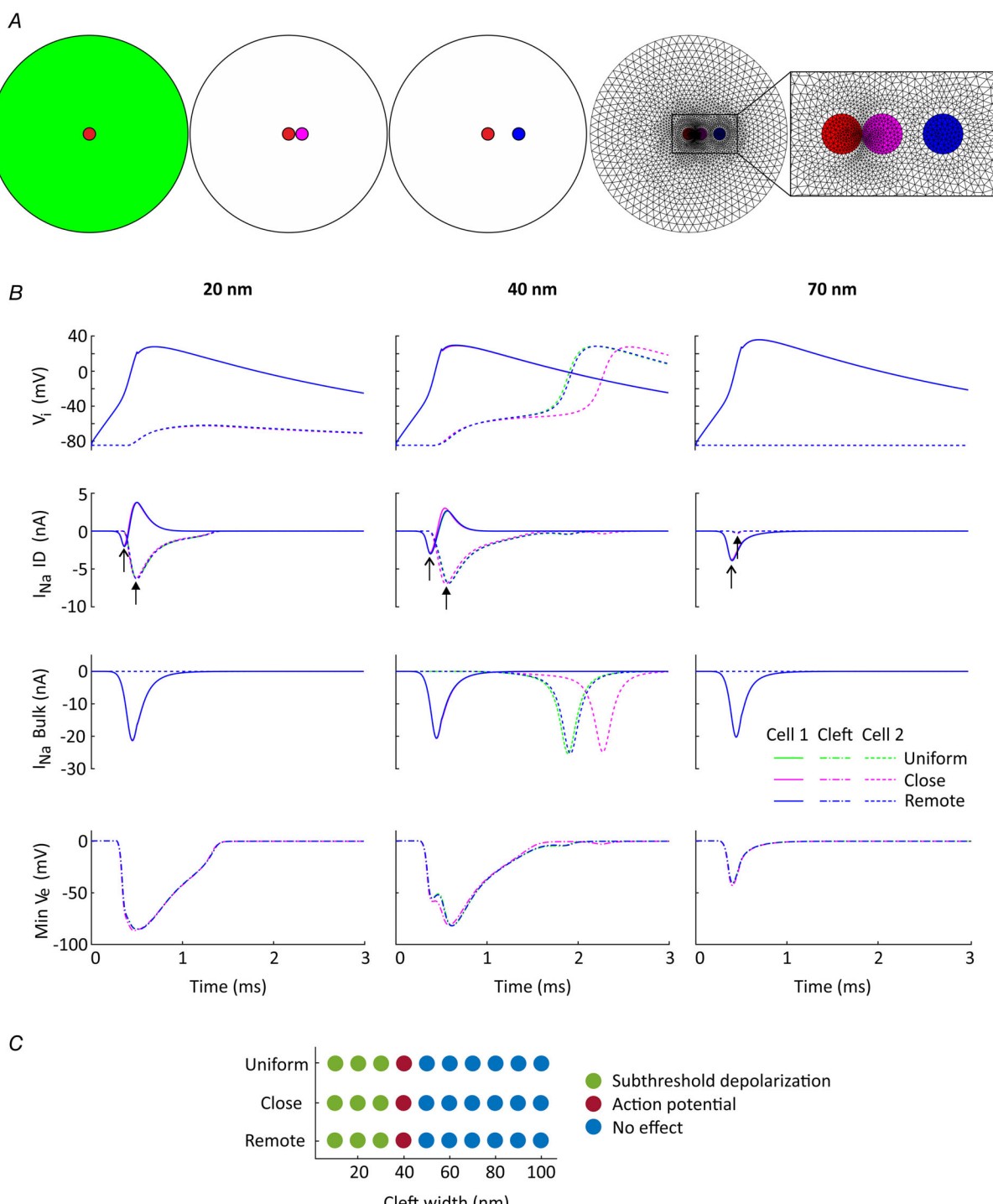

**Figure 3. Effects on AP transmission of different configurations of Na$^+$ channel clusters and gap junctions, for $G_{gap}$ = 0 nS**

*A*, schematic representation and mesh of three ID configurations. From left to right: uniformly distributed gap junctional connection (green) around a centred Na$^+$ channel cluster (red); gap junction plaque (magenta) located close to the centred Na$^+$ channel cluster; gap junction plaque (blue) located more remotely from the Na$^+$ channel cluster; common mesh used for the three configurations with inset showing the region of high element density. *B*, simulation results for zero gap junctional coupling ($G_{gap}$ = 0 nS) and three cleft widths: 20 nm (left column), 40 nm (middle column) and 70 nm (right column). Note that in the absence of gap junctional coupling, the observed differences between the three configurations are due to differences in the conductivities of the extracellular space within the cleft. First row: intracellular potential in cell 1 (continuous lines) and cell 2 (dashed lines); second and third rows: Na$^+$ current of the ID and the bulk membrane for cell 1 (continuous lines) and cell 2 (dashed lines),

respectively; fourth row: minimal extracellular potential in the cleft (dash-dotted lines) as a function of time. Open and closed arrowheads mark the negative peaks of $I_{Na}$ in the ID membranes of cells 1 and 2, respectively. *C*, effects observed on the post-junctional cell for all three ID configurations and for cleft widths varied from 10 to 100 nm.

ID configurations. Of note, in cell 2, depolarization to threshold was due to the ephaptically activated $I_{Na}$ in the post-junctional ID membrane, while the AP upstroke was subsequently driven by bulk $I_{Na}$. It is also worth noting that when $I_{Na}$ in the ID membrane of cell 2 was successfully excited by ephaptic coupling, $I_{Na}$ in the ID of cell 1 became outward because $V_e$, by becoming very negative (around −80 mV), rendered the sign of the $I_{Na}$ driving force $V_m − E_{Na} = V_i − V_e − E_{Na}$ positive.

As illustrated in Fig. 3*C*, for a given cleft width, all three ID configurations resulted in the same global behaviour of cell 2. AP transmission occurred only for a 40 nm wide cleft, whereas narrower clefts led only to sub-threshold depolarization and wider clefts did not lead to any effect on the potential of cell 2. These results indicate that for non-permeable gap junctions, AP transmission does not depend on the co-localization of Na$^+$ channel clusters with gap junction plaques, but rather on ID cleft width.

Results for a low but non-zero $G_{gap}$ of 25.3 nS (1% of normal) are shown in Fig. 4*A*. At this level of coupling, the second cell was always excited because this low level of coupling was sufficient to ensure electrotonic coupling of the two cells, irrespectively of ID configuration. However,

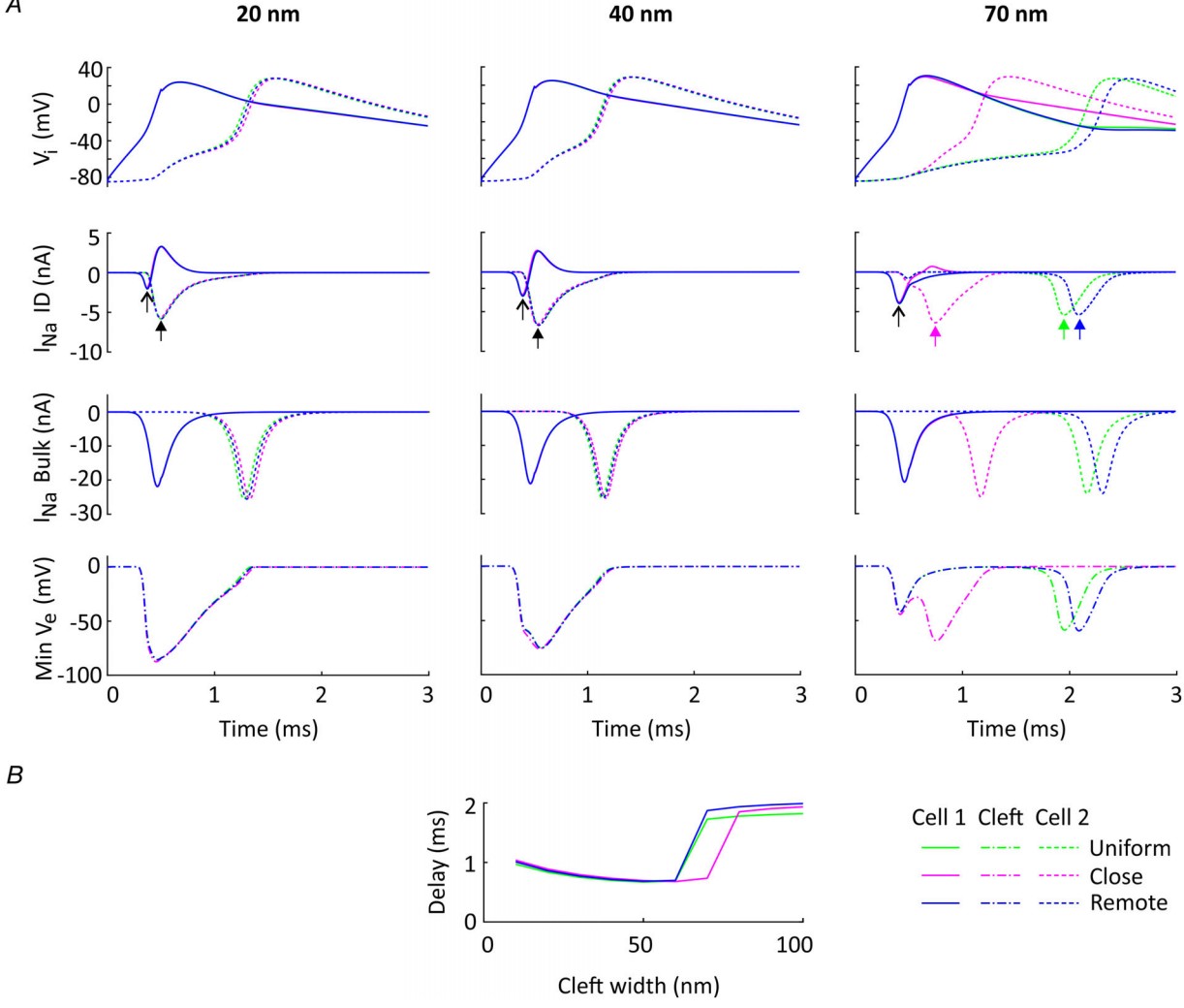

**Figure 4. Effects on AP transmission of different configurations of Na$^+$ channel clusters and gap junctions, for $G_{gap}$ = 25.3 nS (1% of normal)**
*A*, same configurations, simulations and similar figure layout as in Fig. 3*B*, but for a non-zero gap junctional coupling level of $G_{gap}$ = 25.3 nS (1% of normal). *B*, delay between the AP upstrokes in cells 1 and 2 (measured at $V_i$ = 0 mV) as a function of cleft width, for the three ID configurations.

ephaptic coupling modulated the delay between the two AP upstrokes. As detailed in Fig. 4$B$, this delay amounted to approximately 2 ms for cleft widths $\geq$ 80 nm and decreased abruptly to < 1 ms when cleft width was reduced to $\leq$ 60 nm, reflecting the involvement of ephaptic coupling. Except for a 70 nm wide cleft, all results were similar for the three ID configurations (uniform, close and remote). For a 70 nm wide cleft, in comparison to the reference ID (uniform distribution of gap junctional conductance in the ID), the delay between the upstrokes in cells 1 and 2 was slightly longer for the configuration with the gap junction plaque situated remote to the Na$^+$ channel cluster. In contrast, the delay was considerably shorter for the configuration with the closely situated plaque. This short delay was the consequence of the early ephaptic activation of $I_{Na}$ in the ID of cell 2 suggesting that a gap junction plaque closely associated to a Na$^+$ channel cluster may indeed potentiate ephaptic AP transmission in the presence of weak gap junctional coupling. However, this observation is limited to the case of a 70 nm cleft width.

The same simulations were performed with a $G_{gap}$ of 126.5 nS (5% of normal, Appendix Fig. A1). Because this coupling level was now sufficient to ensure rapid electrotonic depolarization of the second cell, the AP was transmitted with almost the same short delay (< 0.5 ms) from the first cell to the second, irrespective of the ID configuration and for all cleft widths. Thus, it was not modulated by ephaptic interactions.

To understand and interpret these results in the context of the two hypotheses presented in Fig. 2, a close examination of the intracellular potentials ($V_i$) on the cytoplasmic sides of the ID membranes is required. Figure 5 shows these intracellular potentials along the ID diameter passing through the Na$^+$ channel cluster and the gap junction plaque. These potentials are represented at the times at which the Na$^+$ current in the ID membranes was maximal (i.e. at the moment of the negative peaks of $I_{Na}$ in the ID of cells 1 and 2, respectively, as marked by arrows in Figs 3$B$, 4$A$ and Appendix Fig. A1). At the time when $I_{Na}$ peaked in the ID membrane of cell 1, $I_{Na}$ was passing from the extracellular cleft through the open Na$^+$ channels into the intracellular space of cell 1. Thus, considered from the intracellular side of cell 1, the Na$^+$ channel cluster represented a current source.

For non-permeable gap junctions ($G_{gap} = 0$ nS, Fig. 5$A$), at the moment $I_{Na}$ peaked in the ID of cell 1, the intracellular side of its ID was already partially depolarized and $V_i$ was slightly more positive by a few millivolts in the proximity of the Na$^+$ channel cluster, reflecting its effect as a current source. However, no change in $V_i$ had yet occurred in cell 2 ($V_i$ remained spatially constant at a resting potential of $-85$ mV). At the instant $I_{Na}$ peaked in the post-junctional ID membrane, for a 70 nm wide cleft, only minimal changes in $V_i$ were observed for cell

2 along its ID diameter because $I_{Na}$ in the post-junctional membrane was too weak (see Fig. 3$B$). In contrast, for cleft widths of 20 and 40 nm, the intracellular side of the ID of cell 2 exhibited a local depolarization by approximately 10 mV in the vicinity of the Na$^+$ channel cluster, again reflecting the ephaptically activated $I_{Na}$. Interestingly, for these two cleft widths, $I_{Na}$ was then flowing out of cell 1 into the intercellular cleft, thus partially contributing to $I_{Na}$ flowing into cell 2 (as marked by open arrows in Fig. 5$A$). This outward $I_{Na}$ manifested as a slightly more negative $V_i$ in the proximity of the Na$^+$ channel cluster because, considered from the intracellular side of cell 1, it represented a current sink. Because there was no gap junctional communication, no current could pass directly from cell 1 into cell 2 to contribute to the activation of Na$^+$ channels in the post-junctional ID membrane. Thus, after entering cell 1 via open Na$^+$ channels on the pre-junctional ID membrane, the charge carried by $I_{Na}$ distributed over the bulk membrane of cell 1 as proposed in Fig. 2$B$. Therefore, the different delays between the AP of cell 1 and 2 observed for a 40 nm cleft in Fig. 3$B$ resulted exclusively from the modified extracellular properties in the cleft due to the presence and the position of the gap junction plaque.

Intracellular potentials along the ID diameters with a $G_{gap}$ of 25.3 nS (1% of normal) are shown in Fig. 5$B$. When $I_{Na}$ peaked in the ID membrane of cell 1, the Na$^+$ channel cluster served again as a current source, but current was now also leaving the intracellular space of cell 1 through the close or remote gap junction plaque (as marked by closed arrows in Fig. 5$B$), which behaved as a current sink when considered from the intracellular side of cell 1 and as a source when considered from the intracellular side of cell 2. In cell 1, this gap junctional sink led locally to a slightly more negative $V_i$ and, in cell 2, to a slightly more positive $V_i$. At the instant of peak $I_{Na}$ in the ID of cell 2, $I_{Na}$ was flowing outwards through the ID membrane of cell 1 and inwards through the ID membrane of cell 2 (see Fig. 4$A$ and open arrows in Fig. 5$B$). In addition, electrotonic current was flowing through the closely or remotely positioned gap junction plaque (closed arrows in Fig. 5$B$). Of note, when the gap junctional conductance was distributed homogeneously around the Na$^+$ channel cluster, electrotonic current did not contribute to local gradients of $V_i$ (not shown). Altogether, the intracellular potential gradients at the level of the ID membranes were small (maximal potential difference < 10 mV) and similar for different locations of the gap junction plaque (close or remote) on the ID. These results indicate that a close Na$^+$ channel cluster-gap junction plaque arrangement does not additionally support ephaptic transmission for low ($G_{gap} = 25.3$ nS, 1% of normal) gap junctional coupling levels via different patterns of intracellular current flow. In the same manner as for a $G_{gap}$ of 0 nS, the differences in AP delay between both adjacent cells (see e.g. 70 nm wide

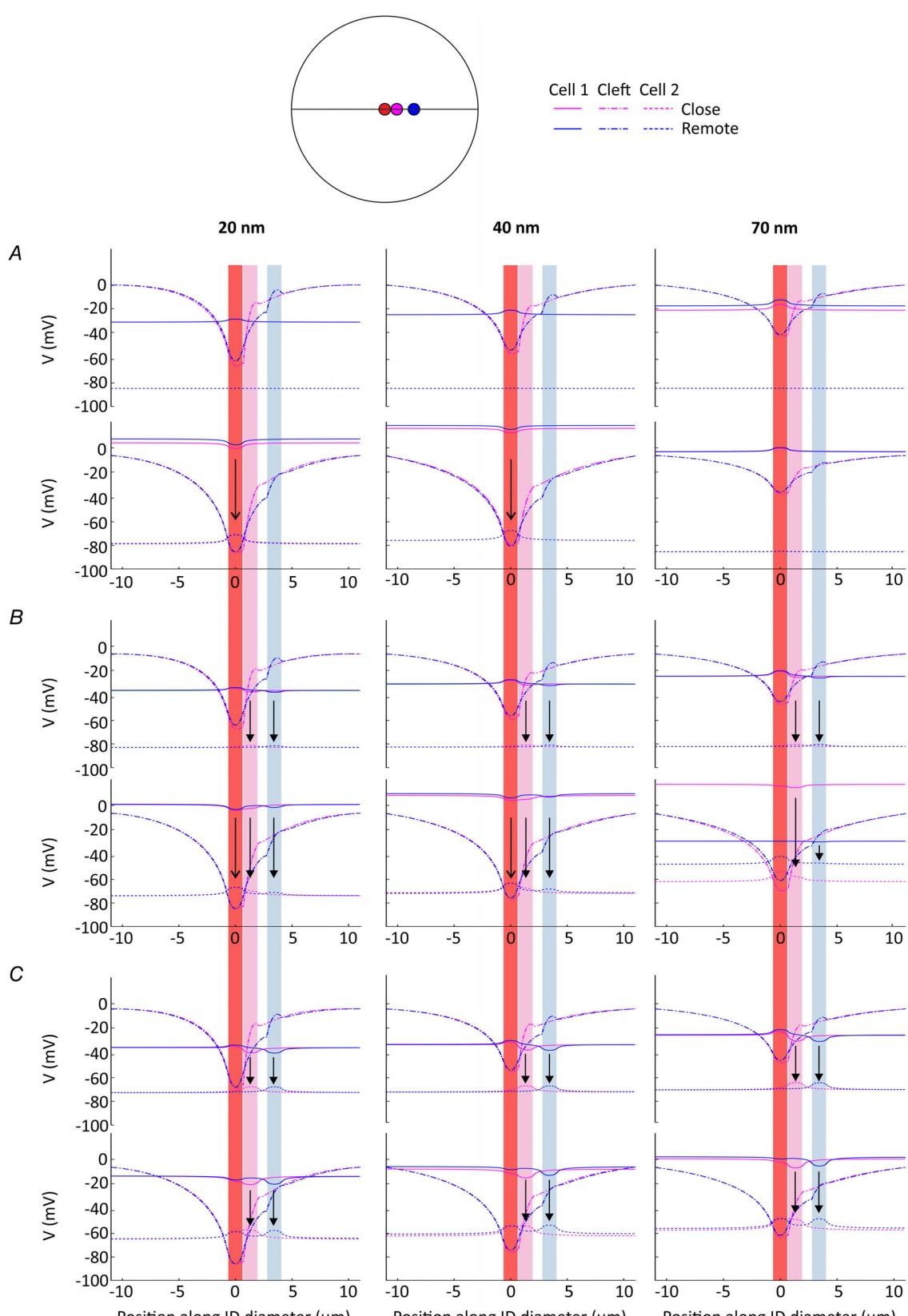

**Figure 5. Spatial profiles of the intracellular potential at the level of the ID**
Extracellular potentials (dash-dotted lines) and intracellular potentials along the ID diameter (see schematic) of cell 1 (continuous lines) and cell 2 (dashed lines) at the instants of peak $I_{Na}$ in the ID of cell 1 (top rows in each panel)

and cell 2 (bottom row in each panel), respectively. *A*, corresponds to simulations in Fig. 3 with $G_{gap} = 0$ nS. *B*, corresponds to simulations in Fig. 4 with $G_{gap} = 25.3$ nS (1% of normal). *C*, corresponds to simulations in Appendix Fig. A1 with $G_{gap} = 126.5$ nS (5% of normal). The red bands mark the position of the Na$^+$ channel cluster; the pink and light blue bands mark the position of the close and remote gap junction plaque, respectively. Open and closed arrowheads mark the current flow from cell 1 into cell 2 through the Na$^+$ channel clusters and the gap junction plaque, respectively.

cleft in Fig. 4*A*) resulted from the different properties of the extracellular ID cleft.

With a $G_{gap}$ of 126.5 nS (5% of normal, Fig. 5*C*), a more substantial part of the positive charge carried by the inward Na$^+$ current in cell 1 was then channelled through the gap junction plaques from the first to the second cell (closed arrows in Fig. 5*C*), as reflected by larger local deflections of $V_i$, suggesting a scenario compatible with the hypothesis in Fig. 2*A*. This current flow reveals the consequence of Na$^+$ channel clusters and gap junction plaques co-localization, which would in principle potentiate conduction via ephaptic coupling. However, because the gap junctional conductance of 126.5 nS (5% of normal) was already sufficient to ensure rapid depolarization of cell 2 to threshold (see Appendix Fig. A1), the ephaptic effects were masked and did not lead to manifest differences between the three configurations.

Further insights into the interplay between the currents flowing through the Na$^+$ channel clusters and the gap junctions are provided by Fig. 6, which shows $V_i$ and current streamlines in the distal part of cell 1 and the proximal part of cell 2, at the respective times of peak $I_{Na}$ in the two ID membranes, for the configuration of the closely adjacent gap junction plaque and a cleft width of 40 nm. With a $G_{gap}$ of 25.3 nS (1% of normal, Fig. 6*A*), at the time $I_{Na}$ peaked in the ID membrane of cell 1, some current streamlines folded back from the Na$^+$ channel cluster (source) to the gap junction (sink), but most of $I_{Na}$ was directed towards the bulk of cell 1. Meanwhile, in cell 2, the gap junctional current was directed to the bulk of cell 2; $I_{Na}$ in the ID of cell 2 was only minimally activated and therefore the streamlines passing through the Na$^+$ channel cluster were displaced to the side. This situation is most compatible with Fig. 2*B*. At the time of peak $I_{Na}$ in the ID membrane of cell 2, $I_{Na}$ was flowing outwardly through the Na$^+$ channel cluster of cell 1 and inwardly through the Na$^+$ channel cluster of cell 2. Thus, part of the intercellular current flowed through the gap junction and part through the Na$^+$ channels of the two facing clusters. The situation with a $G_{gap}$ of 126.5 nS (5% of normal, Fig. 6*B*) was different in two aspects. First, at the time of peak $I_{Na}$ in the ID membrane of cell 1, a smaller proportion of $I_{Na}$ streamlines was channelled through the gap junction, while most of the gap junctional current originated from the bulk of cell 1. Second, at the time of peak $I_{Na}$ in the ID membrane of cell 2, a much larger proportion of current passed through the gap junction in comparison to the Na$^+$ channel clusters. Both aspects can be explained by the

larger gap junctional conductance. These results indicate that the scenarios in Fig. 2 are only partly applicable, depending on the level of coupling provided by the gap junction plaque. For $G_{gap} = 25.3$ nS (1% of normal), at which ephaptic interactions modulate AP transmission, the results are compatible with the situation hypothesized in Fig. 2*B*. However, for $G_{gap} = 126.5$ nS (5% of normal), at which rapid AP transmission is ensured without the need for ephaptic interactions, only the post-junctional schematic in Fig. 2*B* is applicable. Of note, in cell 2, we never observed a curling of streamlines from the gap junction to the Na$^+$ channel cluster as hypothesized in Fig. 2*A*.

It is also important to underline that the intracellular potential gradients remained small, arguing against the possibility that substantial local intracellular potential differences affect the propagation of the AP. Moreover, these intracellular potential differences did not increase when the distance between the gap junction plaque and the Na$^+$ channel cluster was decreased (see Fig. 5). The presented results therefore indicate that when cleft width is homogeneous outside gap junctions, the influence of gap junctional distribution over the ID on ephaptic interactions and AP transmission is very limited. Of note, the presence of a gap junction plaque slightly distorted the spatial profile of $V_e$ (Fig. 5), because the plaque represented an obstacle for extracellular current flow in the cleft. This slight distortion may explain the small differences in terms of AP propagation delay observed for a $w_{cleft}$ of 40 nm in Fig. 3 and of 70 nm in Fig. 4.

### Na$^+$ channel clusters in narrow perinexi enhance ephaptic interactions

At the periphery of gap junctions, in regions known as perinexi, the intercellular cleft is typically narrower (Rhett *et al.* 2013; Raisch *et al.* 2018). In addition, Na$^+$ channel clusters are preferentially located in perinexi (Rhett *et al.* 2013; Veeraraghavan *et al.* 2015; Raisch *et al.* 2018). Thus, we next examined the effects of reducing cleft width in the perinexus on ephaptic AP transmission. The same model with the same boundary conditions and parameters was used as in the previous section, with the difference that a perinexus (with a radius $r_{peri} = 3.3r_{GJ}$) was incorporated around the gap junction plaque. Again, the gap junction plaque and the Na$^+$ channel cluster were either located in close proximity or at larger distance to each other. In consequence, the Na$^+$ channel cluster was located either

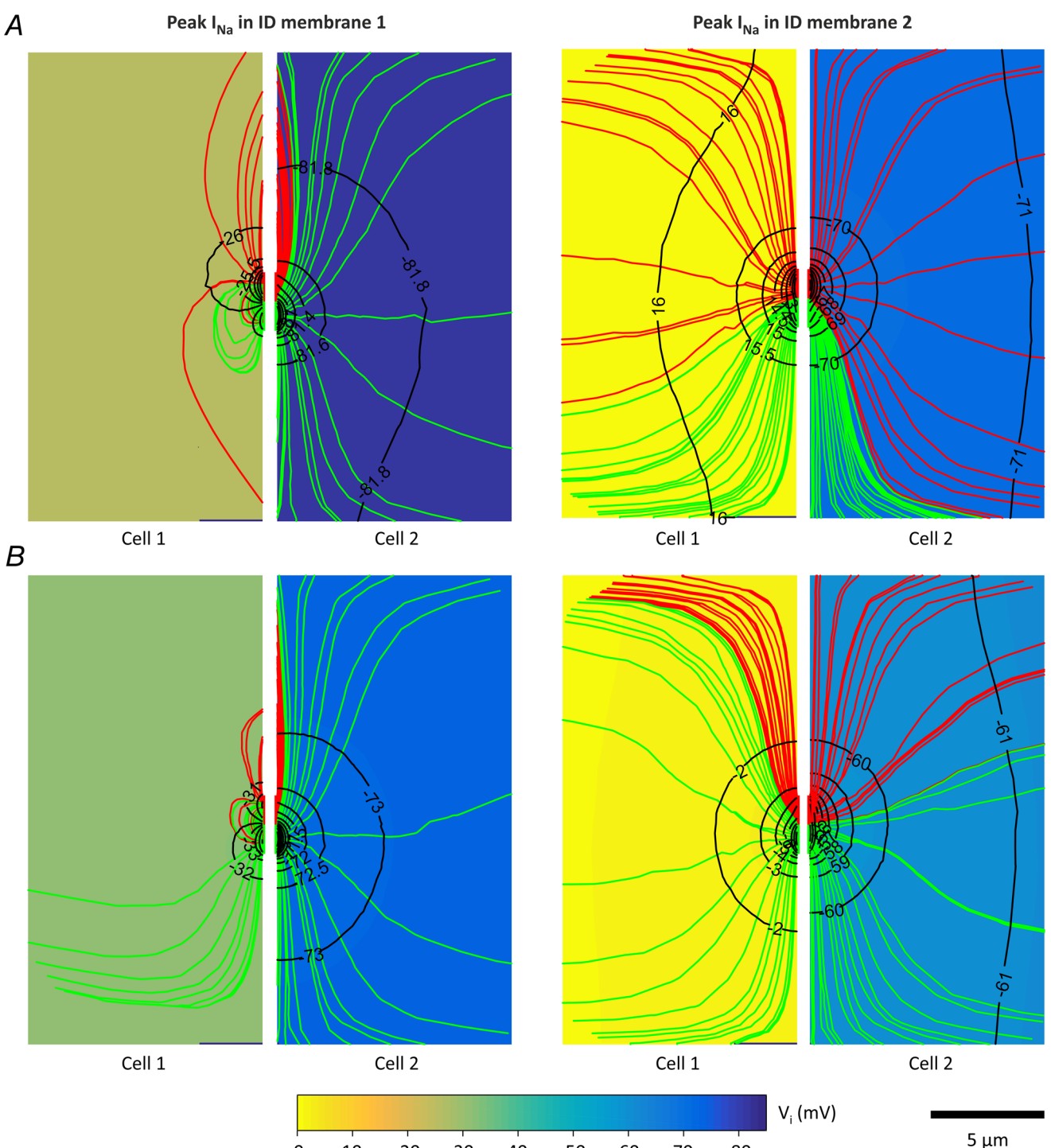

**Figure 6. Intracellular potentials and current streamlines in the cell pair**
Intracellular potentials and current streamlines in cells 1 and 2 in the 2D slice passing through the cell axis, the centres of the $Na^+$ sodium channel cluster and the centre of a closely adjacent gap junction plaque. Isopotential levels are shown in black and current streamlines through the gap junction and the $Na^+$ channel cluster are shown in green and red, respectively. *A*, simulation results with $G_{gap} = 25.3$ nS (1% of normal). *B*, simulation results with $G_{gap} = 126.5$ nS (5% of normal). The ID cleft (40 nm wide) is not shown to scale. Each panel shows potentials and streamlines at the time of the negative $I_{Na}$ peak in the ID membrane of cell 1 (on the left) and cell 2 (on the right), respectively (see Fig. 4*A* and Appendix Fig. A1). The streamlines were constructed by setting their origins at 20 equidistant points placed along the diameters of the $Na^+$ channel cluster (red) and gap junction plaque (green), respectively. The jagged aspect of the streamlines is due to the linear interpolation used in generating this figure.

inside or outside the perinexus, respectively. The intercellular cleft width was varied separately for the region inside ($w_{peri}$) and outside ($w_{cleft}$) of the perinexus, with $w_{peri} \leq w_{cleft}$. We investigated different combinations of bulk cleft widths ($w_{cleft}$ between 10 and 100 nm) and perinexal widths ($w_{peri}$ from 5 nm up to $w_{cleft}$, the latter corresponding to a homogeneous width over the ID) on ephaptic effects.

Figure 7 illustrates the results for the ID configuration in which the $Na^+$ channel cluster was located close to the gap junction plaque and thus inside the perinexus. The gap junctional coupling level was set to zero. Thus, the observed differences with varying $w_{peri}$ are, once again, solely attributable to phenomena arising in the intercellular cleft. For a narrow bulk cleft width of 30 nm, as in the simulations without a narrowed perinexus (Fig. 3), the required threshold to depolarize the entire bulk membrane of cell 2 was not reached, even with $w_{peri} < w_{cleft}$. Moreover, as expected from Fig. 3, bulk cleft widths > 40 nm (e.g. 50 and 100 nm) did not lead to signal transmission when $w_{peri}$ was equal to $w_{cleft}$. However, the presence of a narrowed perinexus in the ID enabled AP propagation for specific combinations of cleft and perinexal width. For example, for a 50 nm wide cleft, the second cell was excited when the perinexal width was set to values between 30 and 40 nm. A further decrease in perinexal width only led to subthreshold depolarization of cell 2. This is because the activated $Na^+$ current in the ID membrane of cell 2 did not suffice to excite the entire second cell. With increasing bulk cleft width, the range of perinexal widths leading to an AP in cell 2 shifted towards smaller values. Indeed, as shown in Fig. 7, for a 100 nm wide cleft, a perinexal width between 20 and 30 nm permitted AP transmission through ephaptic coupling.

For all bulk cleft widths, the minimal extracellular potential became more negative with decreasing perinexal widths. For a 5 nm wide perinexus, the minimal extracellular potential reached a negative value of −100 mV. As shown in our previous computational study, the extracellular potential in the cleft is modulated by ephaptic coupling and is strongly linked to the $Na^+$ current (Hichri *et al.* 2018). Therefore, we examined in detail the $Na^+$ current in the ID membranes of both cells facing one another. Independently of the considered bulk cleft width, we identified four mechanisms modulating the $Na^+$ current. First, the negative peak of the $Na^+$ current in the ID of cell 1 decreased with decreasing $w_{peri}$ because of self-attenuation of the pre-junctional ID $Na^+$ current (arrows labelled 'I' in Fig. 7). Second, self-attenuation also occurred for the post-junctional ID $Na^+$ current (arrows labelled 'II'), thereby decreasing the inward current into cell 2. The self-attenuation of these $Na^+$ currents resulted from the very negative $V_e$, leading to a diminishing driving force ($V_m − E_{Na}$). Third, in contrast, the post-junctional ID $Na^+$ current self-activated, which shortened the

latency of $I_{Na}$ activation in the ID membrane of cell 2 (arrows labelled 'III'). Fourth, we identified a mechanism which we call '$Na^+$ transfer', whereby outward $Na^+$ current through the pre-junctional ID membrane may serve as source of $Na^+$ ions for the inward $Na^+$ current through the post-junctional ID membrane. $Na^+$ current flowing out of cell 1 then contributes, in addition to the cleft space, to the $Na^+$ current flowing into cell 2 (arrows labelled 'IV').

These mechanisms by which perinexal width modulated the ID $I_{Na}$ were only observed when the $Na^+$ channel cluster was located inside the perinexus. Indeed, as shown in Appendix Fig. A2, this modulation of $I_{Na}$ in the ID essentially disappeared when the gap junction plaque and its perinexus were moved away from the $Na^+$ channel cluster, such that the latter was now located outside the perinexal region. This configuration with a remote gap junction plaque only resulted in subthreshold depolarization for a bulk cleft width of 30 nm. For a bulk cleft width of 50 nm, a strongly reduced perinexal width of 5 and 10 nm was required for AP transmission. Larger bulk or perinexal cleft widths led to a complete disruption between both cells, with the $V_i$ of cell 2 remaining near resting potential. These results indicate that in the presence of perinexi with a locally narrower cleft, the relative position between $Na^+$ channel clusters and gap junction plaques is relevant.

Figure 8 shows a comparison of the simulation results for both ID configurations (close and remote gap junction plaques including a perinexus). The gap junctions were non-permeable and various combinations of intercellular widths (inside and outside the perinexus) were analysed for ephaptic effects. Figure 8*A* illustrates the response of the post-junctional cell, categorized as no effect, subthreshold depolarization or AP transmission. The comparison of both ID configurations reveals that more combinations of cleft and perinexal widths led to AP transmission when the $Na^+$ channel cluster was located inside the perinexus. When the gap junction plaque and its perinexus were moved away, the number of combinations as well as the maximal bulk cleft width permitting the excitation of the cell 2 decreased considerably. Indeed, for the remote plaque/perinexus, a bulk cleft width of 50 nm, and only in combination with a maximal perinexal width of 10 nm, was the maximal width for which AP transmission occurred. Further increase of intercellular cleft width led to a total loss of transmission. In comparison, for the ID configuration with the close $Na^+$ channel cluster–gap junction plaque arrangement, AP transmission was possible even for a bulk cleft width of 100 nm when perinexal width was maintained in the range 20–30 nm.

In Fig. 8*B*, the delay between the activation of cell 1 and cell 2 (defined as the time interval between the instants when the $V_i$ upstrokes passed 0 mV) is illustrated

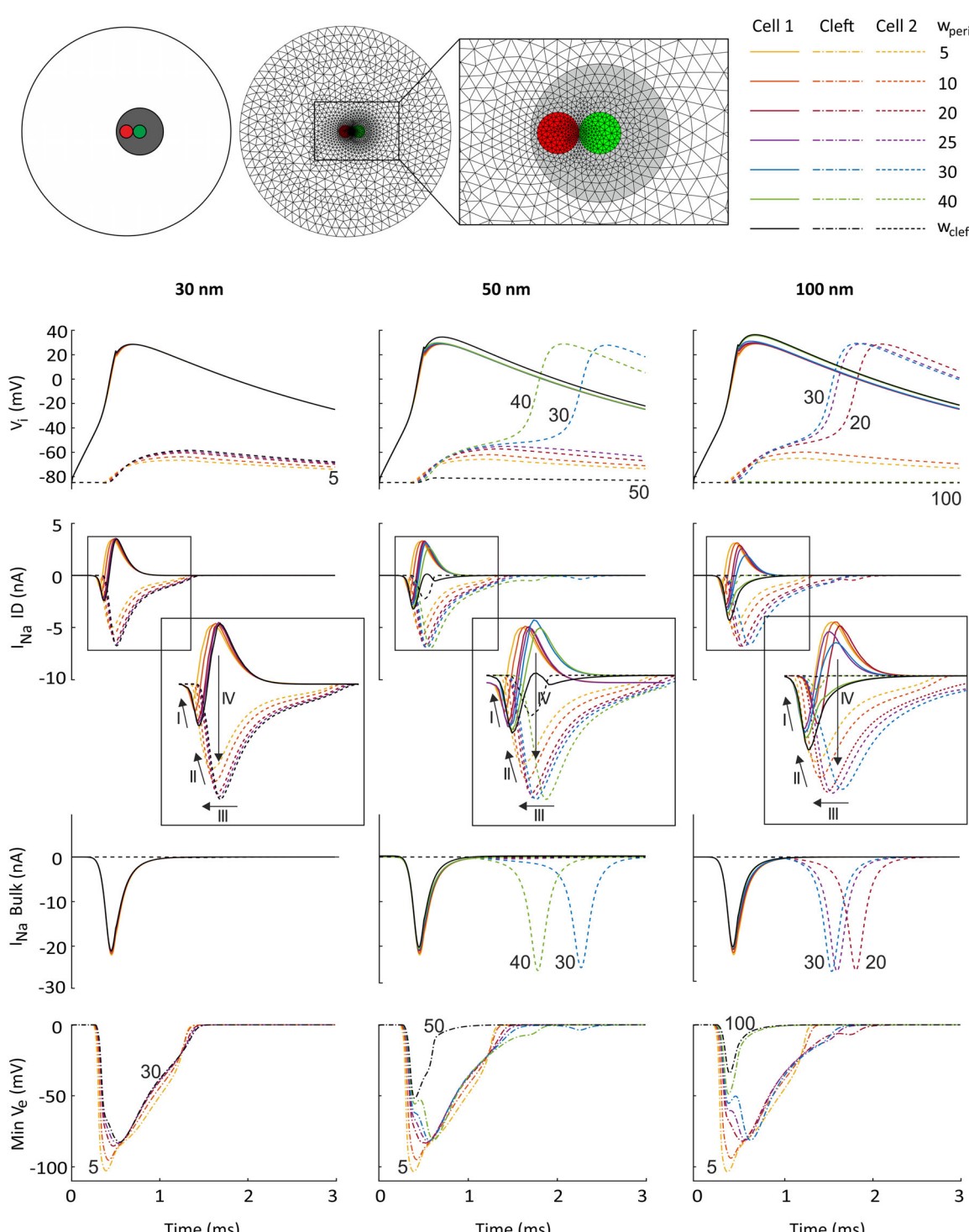

**Figure 7. Effects of perinexal width on ephaptic coupling, for $G_{gap}$ = 0 nS**

Top: schematic and mesh of the ID (inset showing its high density region), with the central Na$^+$ channel cluster (red) located inside the perinexus (grey) of a closely apposed gap junction plaque (green). Simulation results for $G_{gap}$ = 0 nS and bulk cleft widths $w_{cleft}$ of 30 nm (left column), 50 nm (middle column) and 100 nm (right column). Perinexus width $w_{peri}$ was varied from 5 nm up to the bulk cleft width (see colour key upper right). First row: intracellular potential in cell 1 (continuous lines) and cell 2 (dashed lines); second and third rows: Na$^+$ current in the ID and the bulk membrane of cell 1 (continuous lines) and cell 2 (dashed lines), respectively; fourth row: minimal extracellular potential in the cleft (dash-dotted lines) as a function of time. The labels of selected traces indicate $w_{peri}$. The rectangles denote the portions of the plots that were magnified (insets).

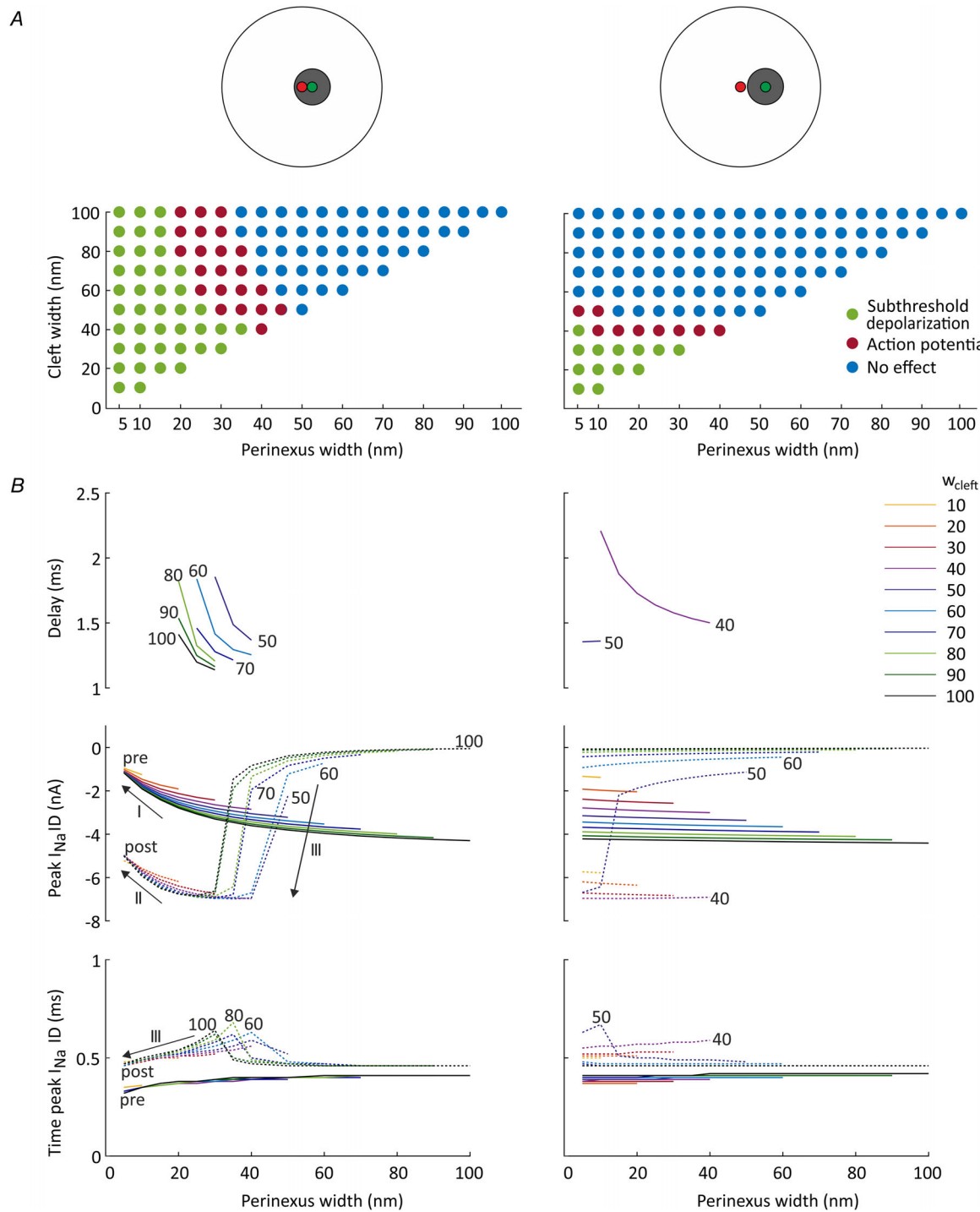

**Figure 8. Comparison of the close and remote localization of the gap junction plaque and its perinexus relative to a central Na⁺ channel cluster in the ID, with $G_{gap} = 0$ nS**

*A*, top: schematic representation of the ID, with the two different positions of the gap junction plaque (green) and its perinexus (grey) relative to the Na⁺ channel cluster (red). The Na⁺ channel cluster is either located inside (left) or outside (right) the perinexus. Bottom: response of the post-junctional cell for various combinations of cleft widths inside and outside the perinexus, categorized as subthreshold depolarization (green), action potential (red) and no effect (blue). Note that the diagonals ($w_{peri} = w_{cleft}$) correspond to simulations presented in Fig. 3*B*. *B*, delay between the activation of the first and the second cell for the two ID configurations (top), negative peak of the Na⁺ current passing the ID membrane of cells 1 and 2 (middle, 'pre' and 'post', continuous and dashed curves, respectively) and corresponding times of the Na⁺ current peaks, as a function of perinexal width (bulk cleft width according to the colour key).

as a function of perinexal width (only combinations of $w_{cleft}$ and $w_{peri}$ leading to successful AP transmission are shown). For $G_{gap} = 0$ nS, a short delay between the two APs was favoured by increasing $w_{peri}$ and decreasing $w_{cleft}$. The width inside the perinexus had a larger effect on the AP delay than bulk cleft width. Figure 8*B* also shows the negative peaks of $I_{Na}$ in the ID membranes of both cells as well as the corresponding times of these $I_{Na}$ peaks as a function of perinexal width. As explained above, self-attenuation in the ID membranes of cells 1 and 2 as well as self-activation in the ID membrane of cell 2 occurred for the close ID configuration (arrows labelled 'I', 'II' and 'III' in Fig. 8*B*), which, altogether, modulated AP transmission and its delay. When the Na$^+$ channel cluster was within the perinexus, for all bulk cleft widths, the magnitude of the post-junctional $I_{Na}$ peak at the ID increased (in absolute value) very steeply when perinexal width was decreased from 100 to 30–50 nm, reaching about $-7$ nA. This abrupt increase of $I_{Na}$ reflects self-activation and ephaptic coupling. Further reduction in perinexal width (30–5 nm), led to self-attenuation of the Na$^+$ currents in the ID membranes of cell 1 and cell 2 and thus, less negative $I_{Na}$ peaks. For narrow perinexi, self-activation of the post-junctional Na$^+$ current led to reduced $I_{Na}$ peak times. This large modulation of ephaptic effects by perinexal width was blunted when the Na$^+$ channel cluster was not located inside the perinexus.

A non-zero $G_{gap}$ of 25.3 nS (1% of normal) was sufficient to permit AP transmission for all combinations of $w_{cleft}$ and $w_{peri}$, but with different AP delays in cell 2, depending on the ID configuration and perinexal width (see Appendix Fig. A3). For the ID configuration with the Na$^+$ channel cluster within the perinexus, the ephaptic modulation of $I_{Na}$ was similar to that with a $G_{gap}$ of 0 nS and a decrease in perinexal width slightly prolonged the delay (see corresponding potentials and currents in Appendix Fig. A4). In contrast, for the ID configuration with the remote gap junctional plaque location, the variation in perinexal width hardly affected the delays between the upstrokes of cells 1 and 2 (see also Appendix Fig. A5). Thus, consistent with the situation with a $G_{gap}$ of 0 nS (Figs. 7 and 8), the same mechanisms (self-attenuation and self-activation) occurred when the gap junctional plaque and its perinexus were close to the Na$^+$ channel cluster, while almost no ephaptic effects occurred with the remote gap junction (Appendix Fig. A3). In the same manner as for zero gap junctional coupling, for a $G_{gap}$ of 25.3 nS (1% of normal), these results highlight the relevance of the localization of Na$^+$ channel clusters in the narrowed perinexi of nearby gap junction plaques. With a $G_{gap}$ of 126.5 nS (5% of normal, not shown), the electrotonic coupling was sufficient to almost synchronize the upstrokes (similar to Appendix Fig A1). However, the modulating effect of perinexal width on $I_{Na}$ self-attenuation and self-activation in the ID membranes

remained when the Na$^+$ channel cluster was located in the perinexus.

To understand the differences arising from the two different relative locations of the Na$^+$ channel cluster and the gap junction plaque with its perinexus, we examined in more detail the extracellular potential and current flow within the ID cleft. Figure 9 illustrates $V_e$ isopotential lines and corresponding current streamlines representing current flow from the periphery of the ID towards the centred Na$^+$ channel cluster in a 50 nm wide cleft, at the time when the minimal $V_e$ was reached. Gap junctional coupling was set to 0 nS. In the presence of a gap junction plaque without narrowed perinexus (Fig. 9A), the current was forced to preferentially flow around it because of the decreased extracellular conductivity in the plaque. This led to a distortion of the streamlines and $V_e$ isopotential lines near the gap junction plaque. However, in the immediate vicinity of the Na$^+$ channel cluster, the spatial pattern of $V_e$ was only minimally affected by the position of the plaque. In contrast, major differences appeared with the additional presence of a 10 nm wide perinexus (Fig. 9*B*). This narrow perinexus, by being *per se* a confined region opposing current flow, resulted in a larger distortion of the $V_e$ isopotential lines and current streamlines. The current diverged to first preferentially flow around the perinexus before converging towards the Na$^+$ channel cluster, in a manner that tended to decrease the path length through the perinexus. Thus, the plaque and its perinexus represented an obstacle for electric current and acted to screen away current flow from the ID periphery into the Na$^+$ channel cluster. This screening effect was considerably more prominent when the Na$^+$ channel cluster was located inside the perinexus, which increased the gradient of $V_e$ (note the higher spatial density of isopotential lines near the Na$^+$ channel cluster in the left panel of Fig. 9*B*) and decreased $V_e$ itself, thereby potentiating ephaptic interactions.

## Discussion

Using our new high-resolution 3D finite element model of two longitudinally abutting cardiomyocytes, we investigated the effect of co-localizing a Na$^+$ channel cluster and a gap junction plaque in the ID on ephaptic transmission in the presence *vs.* absence of a narrowed perinexus. Our first main finding is that in the absence of a narrowed perinexus, the relative position of a gap junction plaque and a Na$^+$ channel cluster only minimally influences ephaptic coupling. This minimal influence is explained by the absence of large spatial gradients of electric potential in the intracellular compartments. In particular, our simulations refute the hypothesis (in Fig. 2*A*) that current flowing through the gap junction exerts a localized depolarization of the post-junctional

membrane which would preferentially activate $Na^+$ channels clustered nearby. Our second main finding is that ephaptic AP transmission is greatly influenced by the localization of $Na^+$ channel clusters within a narrowed perinexus and modulated by perinexal width. This was revealed by the exploration and analysis of the $w_{cleft} - w_{peri}$ parameter space, which showed that spatially heterogeneous cleft widths are crucial for ephaptic coupling when gap junctional coupling is reduced. In the absence of gap junctional coupling (but in the presence of a perinexus), there were specific ranges of $w_{peri}$ that permitted AP transmission even with a large $w_{cleft}$, whereas for homogeneous intercellular cleft widths ($w_{peri} = w_{cleft}$), ephaptic effects mostly vanished. Importantly, this modulation of ephaptic coupling was due to the alteration of current flow within the heterogeneously wide extracellular cleft, which ultimately reinforced the negative $V_e$ at the level of the $Na^+$ channel

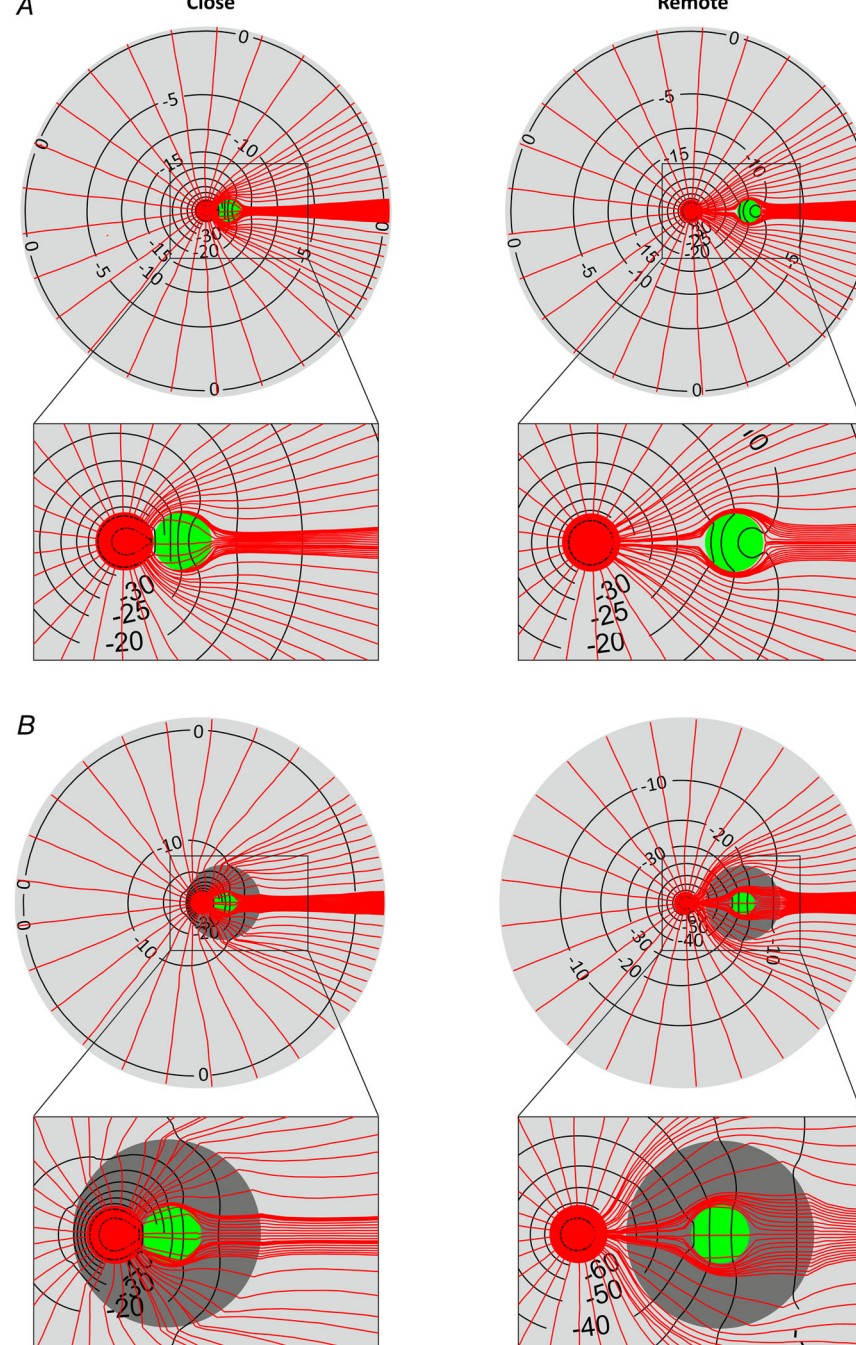

**Figure 9. Effects of the gap junction plaque and perinexus on potentials and current flow in the extracellular cleft**
Extracellular potential (isopotential levels, black, labels in mV) and current flow streamlines (red) in a cleft with $w_{cleft} = 50$ nm, for zero gap junctional coupling ($G_{gap} = 0$ nS). Simulation results at the instants at which the extracellular potential in the cleft was minimal. *A*, ID with a $Na^+$ channel cluster (red) and a close (left) and remote (right) gap junction plaque (green) without a narrowed perinexus ($w_{peri} = w_{cleft} = 50$ nm). *B*, same as *A*, but with a narrowed perinexus ($w_{peri} = 10$ nm, dark grey). The streamlines were constructed with origins placed at predefined positions at the border of the ID.

clusters. A third mechanistic finding of importance is that the direction of $Na^+$ current flow through a $Na^+$ channel cluster in the pre-junctional membrane switches from inward to outward, which may then provide a source of $Na^+$ ions for activating $Na^+$ channels on the post-junctional membrane. Note that this $Na^+$ transfer from one intracellular space to the next may also occur under normal gap junctional coupling conditions, thereby pointing out a further role of the perinexus. This mechanism relies on the presence of $Na^+$ channel clusters that face each other across the perinexal cleft, which has, for the moment, not been demonstrated by morphological studies. Nevertheless, as proposed by Veeraraghavan *et al.* (2018), $\beta 1$ subunits of cardiac $Na^+$ channels may act, via their adhesion function, as scaffold proteins arranging $Na^+$ channels face-to-face across the perinexal cleft.

### Comparison with previous models and studies

In homogeneous models, which are frequently used to represent cardiac cells and cardiac tissue, the geometrical framework is modelled as one single domain, with $V_m$ as variable of interest (Sundnes *et al.* 2006). In the bidomain formulation, the intracellular and extracellular potentials are represented separately, but the intra- and extracellular spaces are still considered to co-exist in every point in space (Tung, 1978; Sundnes *et al.* 2006). However, the precise understanding of the consequences of specific ion channel distributions on the cell membrane requires models based on the geometry of the cell, in which the extracellular space, the intracellular space and the cell membrane are represented explicitly. The cell representation itself in the model then offers the advantage of changing local membrane properties (e.g. $Na^+$ channel density). Tveito and Horgmo Jæger *et al.* developed such a cell-based model (Tveito *et al.* 2017), called the EMI model (extracellular-membrane-intracellular), with non-uniformly distributed $Na^+$ channels on the cell membranes (Horgmo Jæger *et al.* 2019). However, in their model, the finite difference method was applied, limiting the simulation of complex geometries (Horgmo Jæger *et al.* 2021). Our 3D finite element model is based on the work of Agudelo-Toro & Neef (2013) and our own prior 2D model (Hichri *et al.* 2018). The extension from the 2D finite element model of the IDs to a 3D representation of entire cardiomyocytes permitted the consideration of spatially heterogeneous intracellular potentials and thus the representation of localized gap junction plaques in addition to localized $Na^+$ channels in the IDs. Moreover, by refining the numerical methods proposed by Agudelo-Toro and Neef, we were able to reformulate the finite element problem as an easily solvable system with sparse symmetric positive definite matrices.

Previous computational studies (Mori *et al.* 2008; Hand & Peskin, 2010; Lin & Keener, 2010; Tsumoto *et al.* 2011; Veeraraghavan *et al.* 2015, 2016; Wei *et al.* 2016; Greer-Short *et al.* 2017; Weinberg, 2017; Horgmo Jæger *et al.* 2019; Moise *et al.* 2021), including our own (Kucera *et al.* 2002; Hichri *et al.* 2018), have provided important insights into ephaptic coupling as a cardiac conduction mechanism complementing the classical electrotonic mechanism based on current flow via gap junctions. These studies have shown that the high density of $Na^+$ channels in the ID, narrow intercellular cleft widths and $Na^+$ channel clustering are all crucial factors for ephaptic transmission when gap junctional coupling is reduced. Furthermore, based on experimental studies, Poelzing and Gourdie's groups have proposed that perinexi are involved in ephaptic conduction through possible interactions between $Na^+$ channels and gap junctions (Rhett & Gourdie, 2012; Rhett *et al.* 2013; Veeraraghavan *et al.* 2018). To our knowledge, the present study is the first to investigate the functional electrophysiological relevance of $Na^+$ channel clusters being located inside perinexi. Our simulations show that ephaptic coupling is not potentiated by an interaction between $Na^+$ channels and gap junctions, but rather by the confined extracellular space, i.e. by the gap junction plaques themselves (where the membrane spacing is in the range of 2 nm) and mainly by narrowed perinexi. This confinement leads to a screening effect which redirects extracellular current flow and thus enhances ephaptic coupling. Importantly, the range of perinexal widths leading to AP transmission is in line with previous experimental results (Veeraraghavan *et al.* 2015, 2018; King *et al.* 2021).

### Physiological and translational relevance

The physiological and clinical relevance of our findings is underlined by the facts that wider separation between perinexal membranes and disruption of $Na^+$ channel rich ID domains are strongly associated with both atrial and ventricular arrhythmias (Raisch *et al.* 2018; Veeraraghavan *et al.* 2018; Mezache *et al.* 2020). Specifically, Veeraraghavan *et al.* showed that $\beta 1$ subunits of cardiac $Na^+$ channels are also preferentially located in perinexi, where, besides being associated to the channels, they play the role of adhesion molecules spanning the ID cleft and thus maintain its integrity (Veeraraghavan *et al.* 2018). Interestingly, they found that $\beta 1$ subunits are preferentially expressed in $Na^+$ channel clusters located at the edge of gap junction plaques rather than in $Na^+$ channel clusters located next to N-cadherin. Importantly, they observed in guinea-pig ventricles that selective inhibition of $\beta 1$-mediated adhesion with a custom-designed peptide caused perinexal widening and arrhythmogenic slow conduction (Veeraraghavan *et al.* 2018). In another

study, Raisch *et al.* showed that perinexi are wider in patients with a history of atrial fibrillation than in patients without such history (Raisch *et al.* 2018). Similarly, Mezache *et al.* recently reported that exposition of murine hearts to vascular endothelial growth factor A (which is increased in patients suffering from atrial fibrillation) causes acute ID remodelling with perinexal widening as well, accompanied by slowed atrial conduction and an increased propensity to atrial arrhythmias (Mezache *et al.* 2020). Thus, these experimental studies, in combination with our computational results, strongly suggest that alteration of the ID nanostructure, especially peri-nexal widening, leads to conduction disturbances and arrhythmias.

Hence, any approach aimed at preserving (or restoring) the structural integrity of the ID may prove beneficial in cardiology practice and lead, in the future, to new avenues for exploring preventive and therapeutic strategies for cardiac arrhythmias. Such approaches may involve the targeting of $Na^+$ channel $\beta 1$ subunits, N-cadherin or other adhesion molecules. Other approaches may also involve the targeting of intracellular proteins that associate with $Na^+$ channels in the ID and link them to the cytoskeleton, such as ankyrin, SAP97 and plakophilin-2 (Petitprez *et al.* 2011; Shy *et al.* 2013). Thus, new therapeutic options may involve interventions that locally modify intermembrane separation, control the pathological remodelling of the ID nanostructure, or genetic interventions that modulate the expression of structural proteins.

### Limitations and perspectives

High-resolution modelling of cardiac cells involves an increased computational effort. By collapsing the 3D cleft onto a 2D manifold and using different spatial resolutions for the mesh (e.g. very fine mesh for the ID and decreasing mesh density in the extracellular bulk), we were able to limit the computational expense. Because the primary aim of our study was to focus on electrophysiological inter-actions occurring between adjacent cardiomyocytes, we did not investigate conduction in cell fibres or cardiac tissue. Nevertheless, in further steps, our framework can be extended to multicellular structures (albeit at significant computational expense), hopefully providing further insights into ephaptic coupling.

In our model, we assumed a flat ID. It was not in the scope of our study to examine the effects of the known highly tortuous morphology of IDs, as highlighted by studies using super-resolution micro-scopy (Pinali *et al.* 2015; Vanslembrouck *et al.* 2018). Recently, based on transmission electron microscopy images, Moise *et al.* (2021) also discretized IDs using the finite element method to investigate the effect of tortuous and heterogeneous ID structures on conduction. They observed that ID morphology plays an important role

in modulating conduction based on ephaptic coupling. However, in contrast to our work, the high-resolution ID structure was then reduced to regions of different conductivities and incorporated into a simplified electrical network of resistors and capacitors representing cardiac tissue. This permitted the investigation of conduction in a cardiac fibre model using a tractable amount of computational resources, but at the expense of decreased accuracy in the representation of the ID geometry. Notably, the incorporation of $Na^+$ channel clusters is difficult using this approach.

In our high-resolution modelling framework, we focused on the interplay between a gap junction plaque, its perinexus and a $Na^+$ channel cluster located in the ID. Therefore, we removed confounding factors by applying some simplifications to the ID membranes. First, in the ID, all $Na^+$ channels and gap junctions were localized into one single $Na^+$ channel cluster and gap junction plaque, respectively. Second, in our membrane model, we incorporated only $I_{Na}$ and a linear $K^+$ current $I_K$, but not other currents such as the L-type calcium current. However, because we focused only on the AP upstroke, we surmise that the results would not be fundamentally different if we had incorporated currents that are involved in later phases of the AP. Third, we used $Na^+$ and $K^+$ Nernst potentials that were calculated based on constant ion concentrations in the extracellular and intracellular space. Ion concentrations are likely to change in the ID cleft (Mori *et al.* 2008; Greer-Short *et al.* 2017; Nowak *et al.* 2020), which may affect not only ephaptic coupling, but also AP duration (Greer-Short *et al.* 2017; Nowak *et al.* 2020). In a model of ephaptic conduction incorporating electrodiffusion according to the Nernst-Planck equation, Mori *et al.* (2008) have shown that ion concentration changes play a role only for cleft widths in the range of 5 nm. However, they considered a uniform $Na^+$ channel distribution and it is not implausible that the situation may be different in the presence of $Na^+$ channel clusters and narrow perinexi. In particular, $Na^+$ concentration changes may influence the $Na^+$ transfer mechanism that we uncovered in our study.

Although we did not address these complex aspects, our 3D modelling framework can be extended for further investigation of electrotonic and non-electrotonic AP propagation. As a further prospect, it will be important to examine the effects of further distributions of $Na^+$ channel clusters and gap junction plaques with their perinexi in tortuous IDs. We hypothesize that the pre-sence of multiple perinexi and $Na^+$ channel clusters will essentially lead to the same ephaptic mechanisms as we observed for one perinexus. However, tortuosity is expected to strengthen ephaptic mechanisms by increasing the membrane capacitance and the effective radial resistance of the cleft. We also hypothesize that the effects due to the tortuous morphology of the ID in

combination with the position of Na$^+$ channel clusters relative to gap junction plaques and the width of the perinexi will reinforce each other, thus facilitating cardiac conduction via ephaptic coupling when present together rather than separate from each other. Finally, we believe that modelling ion concentration changes as done by Mori *et al.* (2008) is worth exploring, as we expect that these changes will further enhance mechanisms involved in ephaptic coupling, including the Na$^+$ transfer.

## Conclusion

Our 3D finite element model allows a deeper understanding of cardiac conduction with ephaptic inter-

actions. The present study illustrates how modelling and simulation permit the broad investigation of parameter spaces to gain new insights into physiological mechanisms. Our results highlight the importance of Na$^+$ channel clusters being located in regions where the extracellular cleft is locally narrow, as typically in perinexi surrounding gap junction plaques. Importantly, our study demonstrates the role of narrowed perinexi as privileged sites for ephaptic coupling in pathological situations when gap junctional coupling is decreased.

## Appendix

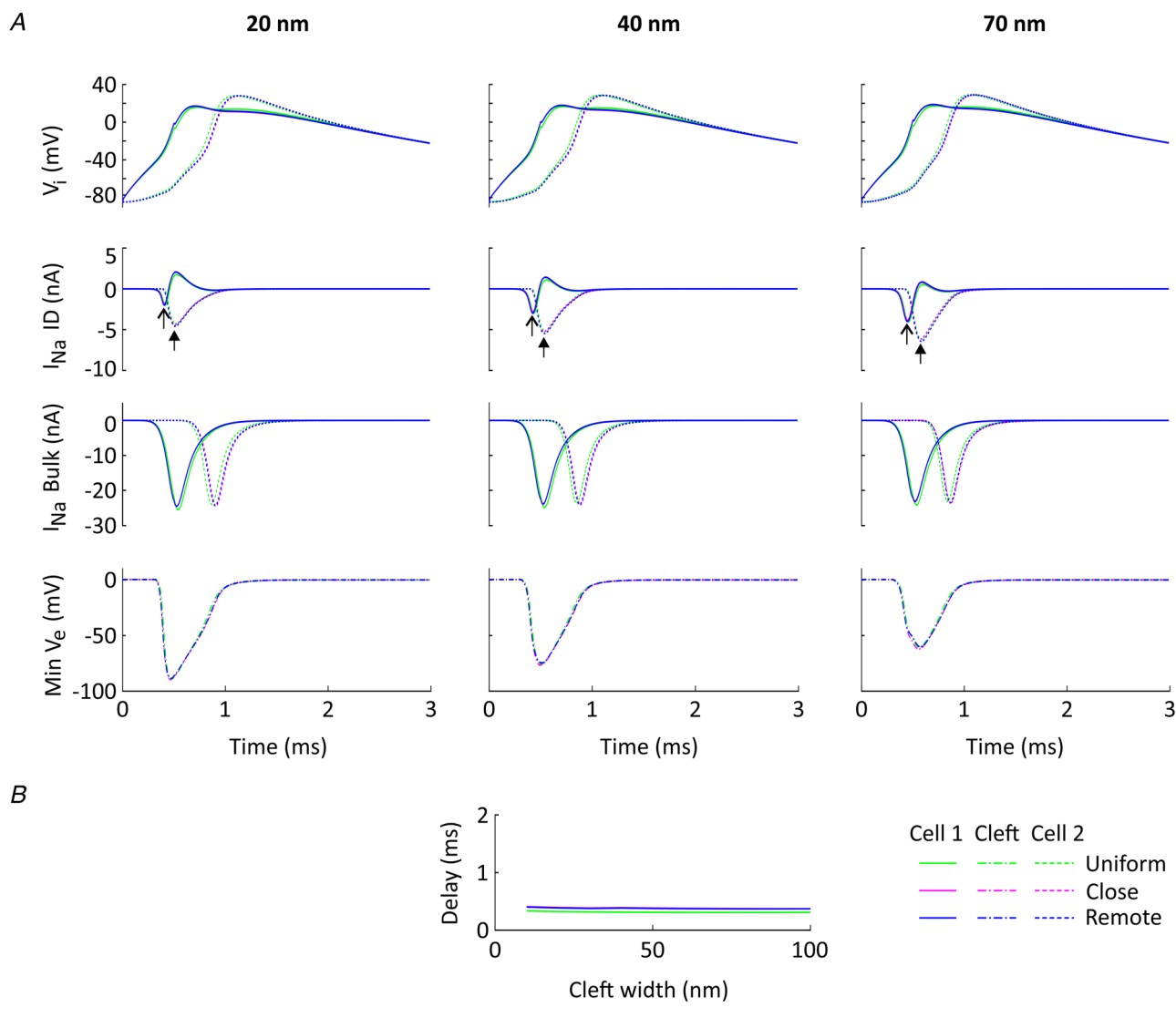

**Figure A1. Effects on AP transmission of different configurations of Na$^+$ channel clusters and gap junctions, for $G_{gap} = 126.5$ nS (5% of normal)**
*A*, same configurations, simulations and (similar) figure layout as in Figs 3*B* and 4*A*, but for a non-zero gap junctional coupling level of $G_{gap} = 126.5$ nS (5% of normal). Note that the curves for the closed (magenta) and remote (blue) gap junction plaque almost overlap. *B*, delay between the upstrokes in cells 1 and 2 as a function of cleft width (compare with Fig. 4*B*).

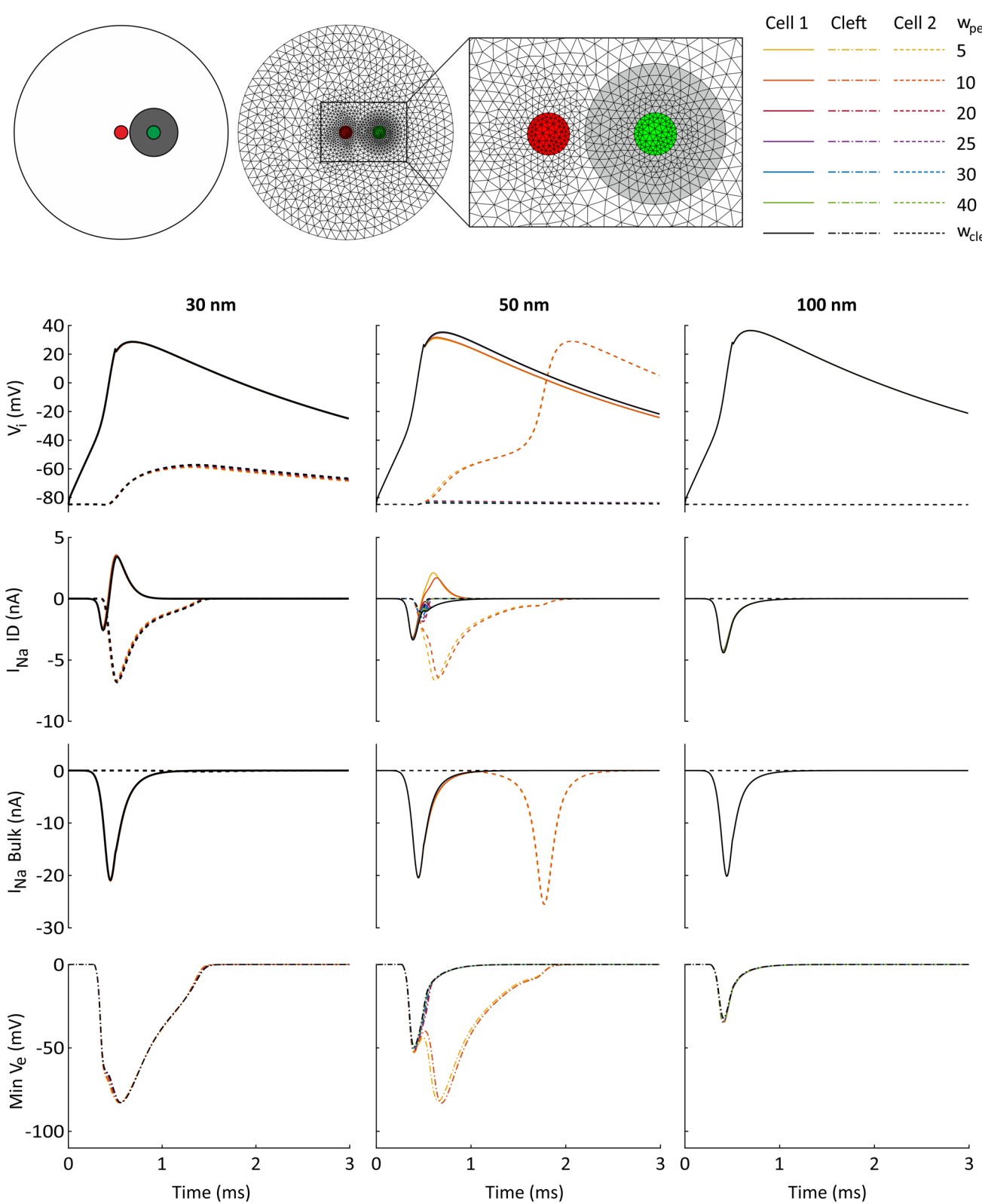

**Figure A2. Effects of perinexal width on ephaptic coupling (Na$^+$ channel cluster outside the perinexus), for G$_{gap}$ = 0 nS**

The gap junction plaque (green) and its perinexus (grey) are located remote from the centred Na$^+$ channel cluster (red), such that the Na$^+$ channel cluster is outside the perinexus (schematic and mesh shown on the top). Same protocol, analysis and layout as in Fig. 7.

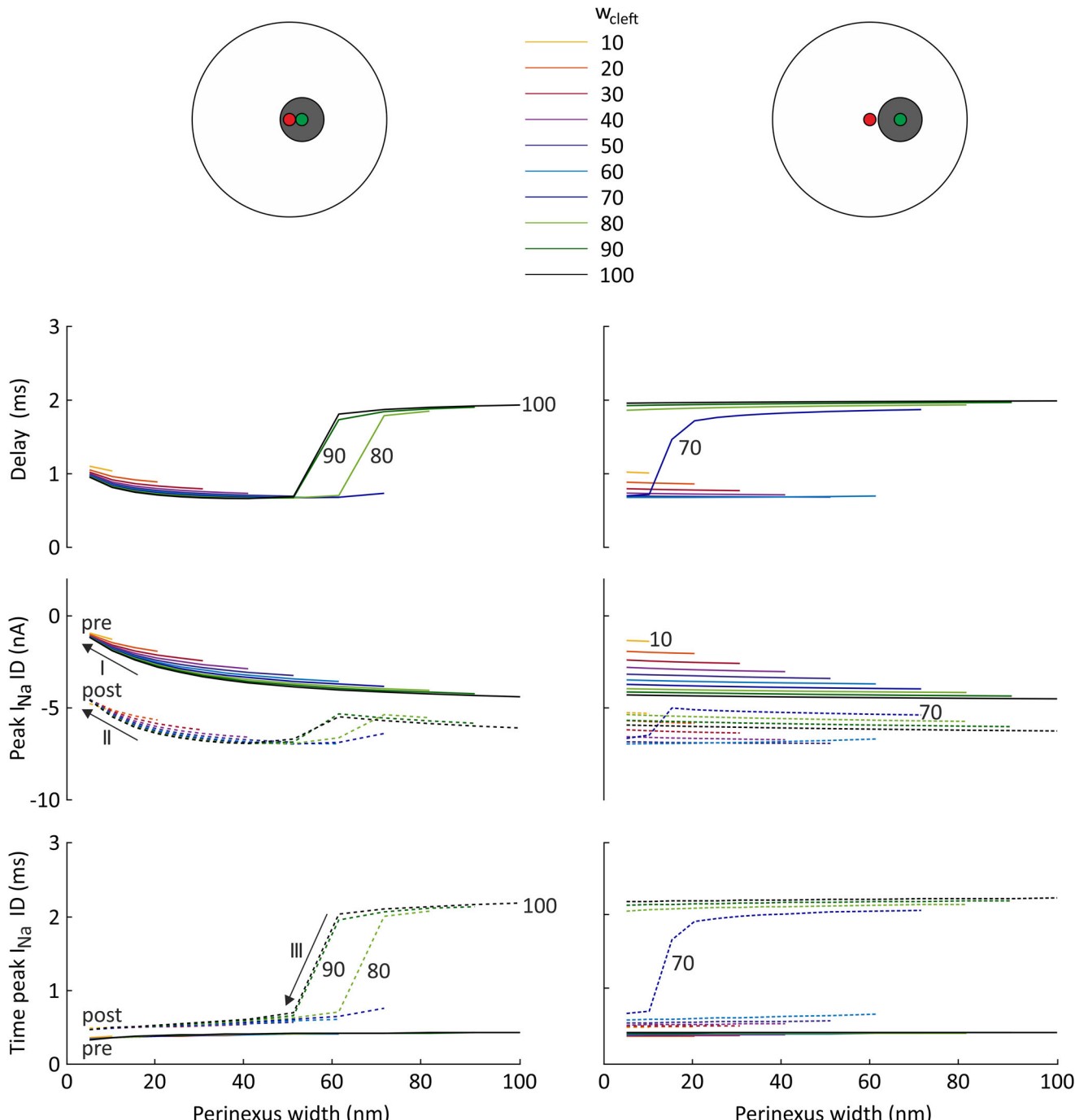

**Figure A3. Comparison of the close and remote localization of the gap junction plaque and its perinexus relative to a central Na$^+$ channel cluster in the ID, with $G_{gap}$ = 25.3 nS (1% of normal)**
Same protocol, analysis and layout as in Fig. 8*B*. Corresponding potentials and currents are shown in Appendix Figs A4 and A5.

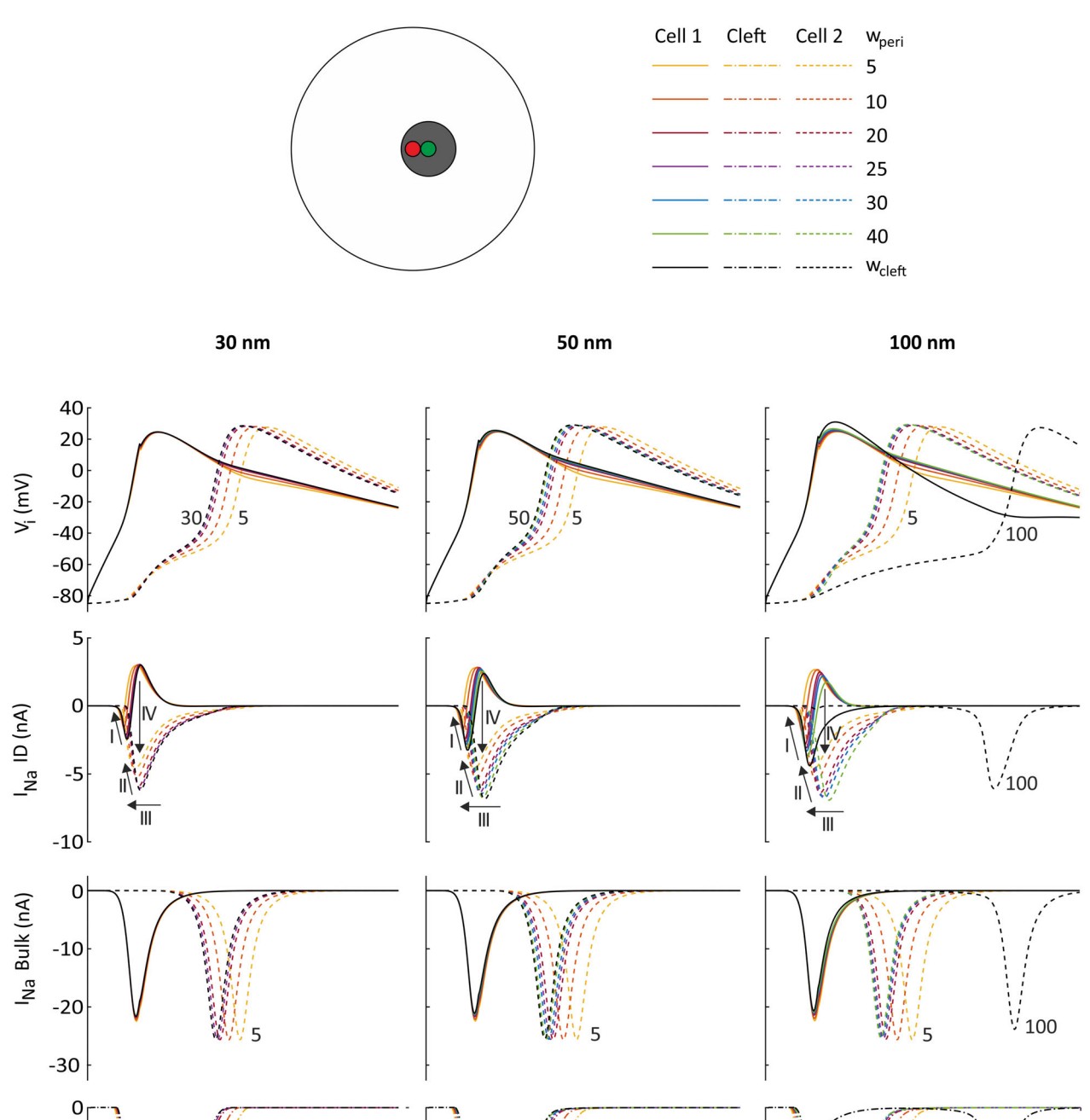

**Figure A4. Effects of perinexal width on ephaptic coupling (Na⁺ channel cluster inside the perinexus), for $G_{gap}$ = 25.3 nS (1% of normal)**

The gap junction plaque (green) and its perinexus (grey) are located close to the centred Na⁺ channel cluster (red), such that the Na⁺ channel cluster is inside the perinexus. Same protocol, analysis and layout as in Fig. 7.

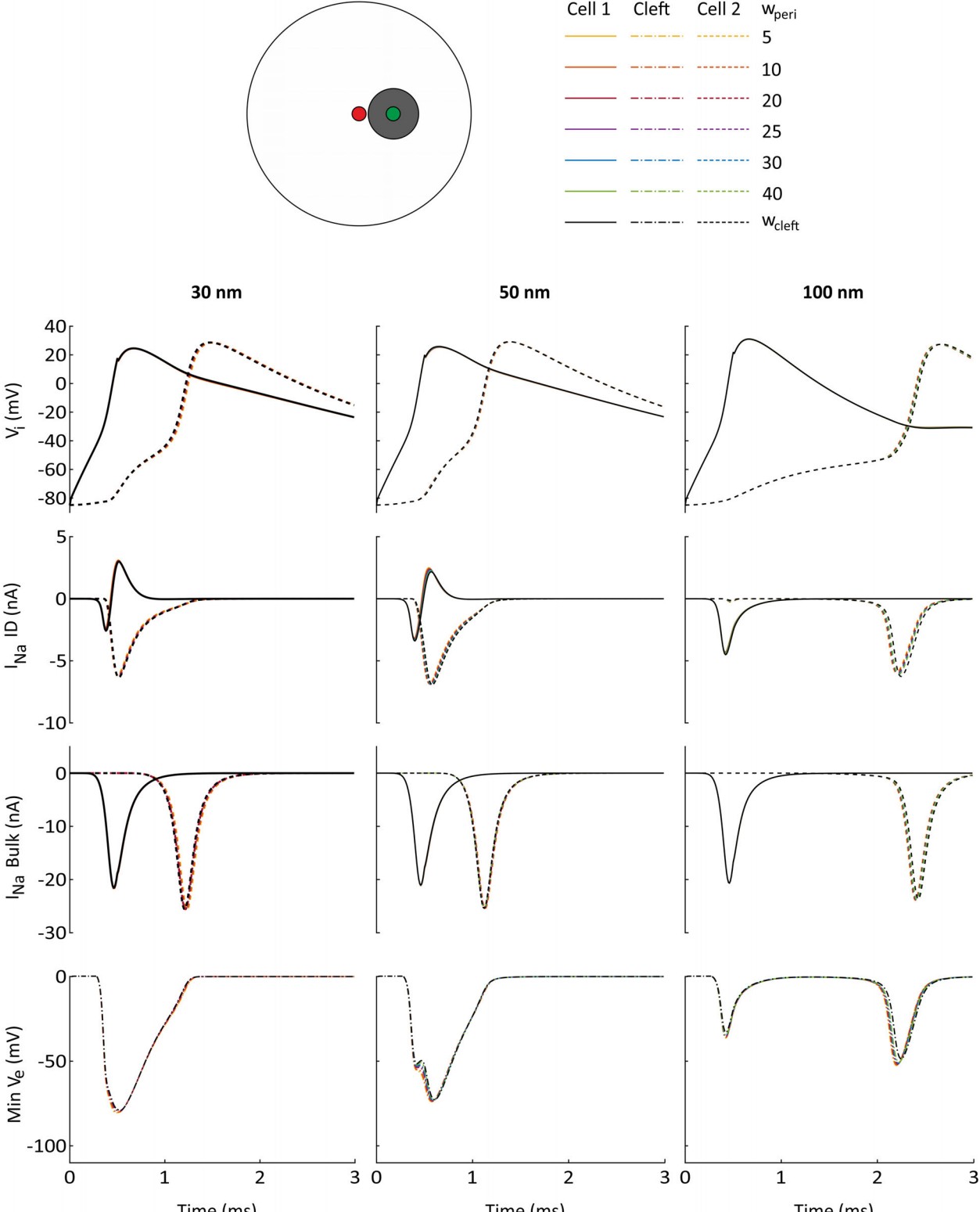

**Figure A5. Effects of perinexal width on ephaptic coupling (Na+ channel cluster outside the perinexus), for $G_{gap}$ = 25.3 nS (1% of normal)**
The gap junction plaque (green) and its perinexus (grey) are located remote from the centred Na+ channel cluster (red), such that the Na+ channel cluster is outside the perinexus. Same protocol, analysis and layout as in Fig. 7.

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

## Additional information

### Data availability statement

The MATLAB code permitting the replication of our simulations is available on the repository Zenodo (http://doi.org/10.5281/zenodo.5226268).

### Competing interests

The authors have no competing interests to disclose.

### Author contributions

J.P.K. and E.I. designed the study. J.P.K. and E.I. developed the numerical methods and the computer code. E.I. conducted the simulations. E.I. prepared the figures. E.I. and J.P.K. drafted and revised the manuscript. Both authors approved the final version of the manuscript, agree to be accountable for all aspects of the work in ensuring that questions related to the accuracy or integrity of any part of the work are appropriately investigated and resolved and both persons designated as authors qualify

for authorship and all those who qualify for authorship are listed.

## Funding

This work was supported by the Swiss National Science Foundation (grant no. 310030_184707 to J.P.K).

## Acknowledgements

The authors are grateful to Steven Poelzing and Robert G. Gourdie for enriching discussions and to Andreas Stahel for his advice on numerical methods.

## Keywords

cardiac electrophysiology, computer modelling, ephaptic coupling, gap junctions, intercalated disc, perinexus, sodium channels

## Supporting information

Additional supporting information can be found online in the Supporting Information section at the end of the HTML view of the article. Supporting information files available:

**Peer Review History**
**Statistical Summary Document**

## Translational perspective

The conduction of the cardiac action potential from cell to cell is essential for the proper function of the heart. Disorders of cardiac conduction occur frequently in the diseased heart and can lead to life-threatening arrhythmias. It has been proposed that ephaptic coupling contributes to cardiac conduction in conjunction with the classical cardiac conduction mechanism based on the current flow through gap junctions. In our modelling study, we found that the clustering of $Na^+$ channels in perinexi surrounding gap junction plaques, together with perinexal width, greatly modulate ephaptic coupling. This finding is relevant in the context of pathologies leading to a change in perinexal width and/or a redistribution of $Na^+$ channels in the intercalated disc. Our study contributes to the general understanding of the physiology and pathophysiology of cardiac conduction at the cellular level, in particular of structure-function relationships in intercalated disc nanodomains and of ephaptic coupling in pathological situations when gap junctional coupling is reduced. The resulting knowledge may, in the future, be a basis to refine diagnostic and therapeutic approaches for cardiac arrhythmias.

