## [Peer Review History · The Journal of Physiology]

□

Localization of Na⁺ channel clusters in narrowed perinexi of gap junctions enhances cardiac impulse transmission via ephaptic coupling: a model study

Ena Ivanovic and Jan P Kucera
DOI: 10.1113/JP282105

Corresponding author(s): Jan Kucera (kucera@pyl.unibe.ch)

The following individual(s) involved in review of this submission have agreed to reveal their identity: Seth Weinberg (Referee #1); Colleen E Clancy (Referee #2)

Review Timeline:	Submission Date:	01-Jul-2021
	Editorial Decision:	23-Jul-2021
	Revision Received:	20-Aug-2021
	Accepted:	06-Sep-2021

Senior Editor: Bjorn Knollmann

Reviewing Editor: Eleonora Grandi

Transaction Report:

Dear Professor Kucera,

Re: JP-RP-2021-282105 "Localization of Na⁺ channel clusters in narrowed perinexi of gap junctions enhances cardiac impulse transmission via ephaptic coupling: a model study" by Ena Ivanovic and Jan P Kucera

Thank you for submitting your manuscript to The Journal of Physiology. It has been assessed by a Reviewing Editor and by 2 expert Referees and I am pleased to tell you that it is considered to be acceptable for publication following satisfactory revision.

The reports are copied at the end of this email. Please address all of the points and incorporate all requested revisions, or explain in your Response to Referees why a change has not been made.

NEW POLICY: In order to improve the transparency of its peer review process The Journal of Physiology publishes online as supporting information the peer review history of all articles accepted for publication. Readers will have access to decision letters, including all Editors' comments and referee reports, for each version of the manuscript and any author responses to peer review comments. Referees can decide whether or not they wish to be named on the peer review history document.

I hope you will find the comments helpful and have no difficulty returning your revisions within 4 weeks.

Your revised manuscript should be submitted online using the links in Author Tasks Link Not Available.

Any image files uploaded with the previous version are retained on the system. Please ensure you replace or remove all files that have been revised.

REVISION CHECKLIST:

- Article file, including any tables and figure legends, must be in an editable format (eg Word)
- Upload each figure as a separate high quality file
- Upload a full Response to Referees, including a response to any Senior and Reviewing Editor Comments;
- Upload a copy of the manuscript with the changes highlighted.

- A potential 'Cover Art' file for consideration as the Issue's cover image;
- Appropriate Supporting Information (Video, audio or data set https://jp.msubmit.net/cgi-bin/main.plex?form_type=display_requirements#supp).

To create your 'Response to Referees' copy all the reports, including any comments from the Senior and Reviewing Editors, into a Word, or similar, file and respond to each point in colour or CAPITALS and upload this when you submit your revision.

I look forward to receiving your revised submission.

If you have any queries please reply to this email and staff will be happy to assist.

Yours sincerely,

Bjorn Knollmann
Senior Editor
The Journal of Physiology

EDITOR COMMENTS

Reviewing Editor:

Both reviewers commented positively on the rigor of the study, novelty, and potential impact of the findings. Areas of improvement include 1) prefacing the detailed methods with some broader guidance for the Journal's readership regarding the overall methodological approach, experimental design, and technical advances; 2) discussing the physiological and translational relevance of the findings; 3) improving clarity by simplifying the information-dense figures or considering alternative ways to present the data.

Senior Editor:

Excellent modeling study that could be improved by simplifying the information-dense figure

REFEREE COMMENTS

Referee #1:

In this study by Ivanovic and Kucera, the authors develop a high-resolution finite-element model of two cells and their shared intercalated disc and specifically represent Na⁺ channel clustering preferentially around gap junctional plaques in the region termed the perinexus. The authors demonstrate the location of these Na⁺ channel clusters near gap junctions and within a narrow perinexus facilitates impulse transmission by ephaptic coupling. The investigation of the mechanism of how opposing Na⁺ channel clusters interact is highly novel, and the numerical methods developed for this computational model are innovative and technically challenging.

This study addresses an important question, as the role of ephaptic coupling is increasingly recognized as an important contributor to cell-cell coupling and electrical conduction. The computational study is technically rigorous, and the manuscript is well written. I have mostly minor comments regarding issues of presentation and discussion.

1. The methods are quite detailed, and normally I would suggest that much of the detail be moved to Supplemental Material. However, as I understand, the journal policies do not allow for this. With that in mind, to make the paper more accessible to the broader readership of the journal, I would suggest that the authors include a brief description early in the Methods that outlines the overall methodology, with the full derivations and equations presented after. This outline could highlight the experimental design and technical advances, without all of the details.

I leave the specifics to the authors' discretion, but some details in the Methods could probably also be shortened or removed. For example, the proof that the matrix presented in Eqn. 1.43 is symmetric and positive definite (while useful) is probably unnecessary for the readership of the journal.

2. A few other Methods questions/comments:

Were all simulations performed using the "collapsed cleft" version of the model? Please clarify and state specifically in the methods. The "Practical implementation" section in the Methods

refers to both the explicit and collapsed cleft models.

Possible typo: Should Eqn. 1.56 have the term " $(1-P)$ " instead of " $(P-1)$ " (although for the particular value of $P = 0.5$ the terms are equivalent)?

3. The authors describe the relative strength of gap junctional coupling as a percentage of some maximal conductance. First, I did not find the value of 100% gap junctional conductance given. But regardless, I would suggest that the authors simply present the value of the conductance for each case, instead of a percentage, in particular in the context that experimental measurements of gap junctional conductance vary quite a lot.

The authors could then provide some context for the relative magnitude of gap junctional coupling. However, considering that physiologically there are many gap junctional plaques between two coupled cells, the "low" conductance values used in the study are potentially very similar to the magnitude of the conductance for a single plaque.

4. Figure 6 is a very information dense figure, and the description of the results for this figure were difficult to follow. I would suggest adding arrows or other symbols with labels that are identified in the main text, similar to other figures, to help clarify for the reader the changes described in the text.

5. In Figure 4, panel A, for the 70 nm column and the Min Ve row, it appears that the Cell 2 lines are plotted with solid lines, not dotted.

Referee #2:

In the paper entitled "Localization of Na^+ channel clusters in narrowed perinexi of gap junctions enhances cardiac impulse transmission via ephaptic coupling: a model study", by Ivanović and Kucera, the authors set out to determine whether clusters of sodium channels near gap junctional plexi may provide sufficient current to promote ephaptic coupling between cardiac myocytes. To address this question, they developed a three dimensional finite element model of two connected cardiac myocytes that included sodium channels clusters on the membranes of the intercalated disk. The model prediction strongly suggest that the location of sodium channels clusters and the width of the intercalated disc are critical variables that promote ephaptic coupling. This is a very well done paper and an interesting topic that has been of interest to the field for some time. The study constitutes the most comprehensive modeling study I've seen to date.

The authors eloquently suggest that it is in the narrow perinexus that comprises a "privileged site" for ephaptic coupling. This may be a particularly relevant in the setting of pathology when

junctional coupling is reduced. This is really a beautifully and clearly written study.

I wonder if the authors might be able to add some more physiological and translational relevance in their discussion. At present, discussion it's a nice summary of the study and technical studies on ephaptic coupling that have come before. However, can the authors elaborate on how understanding the importance of the perinexus localization in promoting ephaptic coupling that seems to be especially relevant in potential disease states? How might having this new information lead to new avenues for exploring therapeutic options? Are there any existing data that suggest a specific proteins complex organization in the ID, and the PN region in particular that would allow those sodium channels to be specifically targeted, for example?

In general, I found the study to be compelling and comprehensive. It is a very good example for where modeling and simulation can allow broad investigation of parameter space related to phenomena. The others might want to mention this as well in the discussion.

I'm not entirely convinced that 13 figures are necessary. Would it be possible to encode some of the data in a table instead of the mini time course data that are included. For me, the big findings and understanding get lost in the many many details reported in the figures. I think many figures could be collapsed into a table or heat map type graphic.

END OF COMMENTS

Confidential Review

01-Jul-2021

Response to Editors and Referees

JP-RP-2021-282105

Localization of Na⁺ channel clusters in narrowed perinexi of gap junctions enhances cardiac impulse transmission via ephaptic coupling: a model study

Ena Ivanovic and Jan P Kucera

We are grateful to the Editors and the Referees for taking the time to review our manuscript and for their constructive comments, which greatly helped us to improve our manuscript. Detailed responses to the specific issues raised and descriptions of the changes made to the manuscript are provided in blue below. In the revised manuscript, our changes are highlighted in yellow.

We have also incorporated a first author profile after the cover page.

We also changed the numbering format for the equations (1 to 55 instead of 1.1 to 1.55).

To comply with the open data policies of our funding agency, we have uploaded the MATLAB source code permitting the replication of our simulations on the repository Zenodo. This is now indicated in the Data Availability Statement. The DOI was not made public yet.

Response to the Reviewing Editor

Both reviewers commented positively on the rigor of the study, novelty, and potential impact of the findings. Areas of improvement include 1) prefacing the detailed methods with some broader guidance for the Journal's readership regarding the overall methodological approach, experimental design, and technical advances; 2) discussing the physiological and translational relevance of the findings; 3) improving clarity by simplifying the information-dense figures or considering alternative ways to present the data.

We thank the Reviewing Editor for appreciating our work. We have followed all three suggestions in the revised manuscript, as detailed below.

Response to the Senior Editor

Excellent modeling study that could be improved by simplifying the information-dense figure.

We thank the Senior Editor for complimenting us on our study. As suggested, we have simplified the information-dense figure, as detailed below.

Response to Referee #1

In this study by Ivanovic and Kucera, the authors develop a high-resolution finite-element model of two cells and their shared intercalated disc and specifically represent Na⁺ channel clustering preferentially around gap junctional plaques in the region termed the perinexus. The authors demonstrate the location of these Na⁺ channel clusters near gap junctions and within a narrow perinexus facilitates impulse transmission by ephaptic coupling. The investigation of the mechanism of how opposing Na⁺ channel clusters interact is highly novel, and the numerical methods developed for this computational model are innovative and technically challenging.

We thank Referee #1 for appreciating our work.

This study addresses an important question, as the role of ephaptic coupling is increasingly recognized as an important contributor to cell-cell coupling and electrical conduction. The computational study is technically rigorous, and the manuscript is well written. I have mostly minor comments regarding issues of presentation and discussion.

We thank the Referee for these encouraging words and for his/her comments, which we have addressed as explained below.

1. The methods are quite detailed, and normally I would suggest that much of the detail be moved to Supplemental Material. However, as I understand, the journal policies do not allow for this. With that in mind, to make the paper more accessible to the broader readership of the journal, I would suggest that the authors include a brief description early in the Methods that outlines the overall methodology, with the full derivations and equations presented after. This outline could highlight the experimental design and technical advances, without all of the details.

Indeed, we agree that the detailed description of our Methods would typically be incorporated into an online Supplemental Material, but, as the Referee correctly points out, this is not the policy of the Journal of Physiology. Therefore, we followed the Referee's suggestion and incorporated an introductory subsection ("General principles") at the beginning of the Methods section outlining the biophysical principles of the model, the methodologies used, the innovative aspect of our approach and an explicit sentence informing that detailed information is presented in the rest of the section, allowing a more general reader to skip this part.

I leave the specifics to the authors' discretion, but some details in the Methods could probably also be shortened or removed. For example, the proof that the matrix presented in Eqn. 1.43 is symmetric and positive definite (while useful) is probably unnecessary for the readership of the journal.

We agree that this proof will probably be unnecessary for the vast majority of readers. We followed the Referee's suggestion and removed the proof and Eqn. 1.43 from the Methods section.

2. A few other Methods questions/comments:

Were all simulations performed using the "collapsed cleft" version of the model? Please clarify and state specifically in the methods. The "Practical implementation" section in the Methods refers to both the explicit and collapsed cleft models.

All simulations presented in our manuscript were performed using the collapsed cleft model. However, both approaches need to be described in the Methods section because the collapsed cleft model builds up on the explicit cleft model (it is both an extension and a simplification). However, in preliminary simulations, we used the explicit cleft model to validate the collapsed cleft model during its development. We clarified this in the new "General principles" subsection and the "Practical implementation" subsection of the Methods.

Possible typo: Should Eqn. 1.56 have the term "(1-P)" instead of "(P-1)" (although for the particular value of $P = 0.5$ the terms are equivalent)?

We thank the Referee for spotting this mistake, which we corrected in the revised manuscript.

3. The authors describe the relative strength of gap junctional coupling as a percentage of some maximal conductance. First, I did not find the value of 100% gap junctional conductance given. But regardless, I would suggest that the authors simply present the value of the conductance for each case, instead of a percentage, in particular in the context that experimental measurements of gap junctional conductance vary quite a lot.

The value corresponding to 100 % overall gap junctional conductance is 2.53 μS . This was and is still stated in the second paragraph of the Methods section. We followed the Referee's suggestion and now, in the Results section and the Figures, we systematically provide the absolute value of this conductance for every case. However, we decided to leave the values as percentages for readers who may be less familiar with absolute values for the intercellular conductance. We also incorporated the value corresponding to 100% in Table 1.

The authors could then provide some context for the relative magnitude of gap junctional coupling. However, considering that physiologically there are many gap junctional plaques between two coupled cells, the "low" conductance values used in the study are potentially very similar to the magnitude of the conductance for a single plaque.

We thank the Referee for this comment. We have added a corresponding note at the end of the second paragraph of the Results section.

4. Figure 6 is a very information dense figure, and the description of the results for this figure were difficult to follow. I would suggest adding arrows or other symbols with labels that are identified in the main text, similar to other figures, to help clarify for the reader the changes described in the text.

To make this Figure (Figure 5 in the revised manuscript) clearer, we removed the data corresponding to the uniform ID configuration (which was not addressed in detail in our initial manuscript and which, in our opinion, bring less useful information) and wrote "not shown" in the corresponding descriptive text in the results section. This Figure now has less quasi-overlapping curves. Furthermore, we followed the Referee's suggestion and added arrows that we identify and describe in the legend and the main text.

5. In Figure 4, panel A, for the 70 nm column and the Min V_e row, it appears that the Cell 2 lines are plotted with solid lines, not dotted.

We thank the Referee for this comment. The extracellular potential in the ID cleft is the same for both cell 1 and cell 2. Thus, min V_e is also the same for both cells. To avoid giving the impression that the plots of V_e or of min V_e pertain to cell 1 only, we changed the continuous line style to a dash-dotted line style for V_e and min V_e in all figures.

Response to Referee #2

In the paper entitled "Localization of Na^+ channel clusters in narrowed perinexi of gap junctions enhances cardiac impulse transmission via ephaptic coupling: a model study", by Ivanović and Kucera, the authors set out to determine whether clusters of sodium channels near gap junctional plexi may provide sufficient current to promote ephaptic coupling between cardiac myocytes. To address this question, they developed a three dimensional finite element model of two connected cardiac myocytes that included sodium channels clusters on the membranes of the intercalated disk. The model prediction strongly suggest that the location of sodium channels clusters and the width of the intercalated disc are critical variables that promote ephaptic coupling. This is a very well done paper and an interesting topic that has been of interest to the field for some time. The study constitutes the most comprehensive modeling study I've seen to date.

We thank Referee #2 for appreciating our study.

The authors eloquently suggest that it is in the narrow perinexus that comprises a "privileged site" for ephaptic coupling. This may be a particularly relevant in the setting of pathology when junctional coupling is reduced. This is really a beautifully and clearly written study.

We thank the Referee for complimenting us on our study.

I wonder if the authors might be able to add some more physiological and translational relevance in their discussion. At present, discussion it's a nice summary of the study and technical studies on ephaptic coupling that have come before. However, can the authors elaborate on how understanding the importance of the perinexus localization in promoting ephaptic coupling that seems to be especially relevant in potential disease states? How might having this new information lead to new avenues for exploring therapeutic options? Are there any existing data that suggest a specific proteins complex organization in the ID, and the PN region in particular that would allow those sodium channels to be specifically targeted, for example?

We thank the Referee for this suggestion. We integrated a new paragraph in the Discussion entitled "Physiological and translational relevance", where we elaborate on these aspects.

In general, I found the study to be compelling and comprehensive. It is a very good example for where modeling and simulation can allow broad investigation of parameter space related to phenomena. The others might want to mention this as well in the discussion.

We thank the Referee for this suggestion. Accordingly, we integrated the sentence "The present study illustrates how modelling and simulation permit the broad investigation of parameter spaces to gain new insights into physiological mechanisms" in our conclusion.

I'm not entirely convinced that 13 figures are necessary. Would it be possible to encode some of the data in a table instead of the mini time course data that are included. For me, the big findings and understanding get lost in the many many details reported in the figures. I think many figures could be collapsed into a table or heat map type graphic.

We have considered the Referee's suggestion to present some data in the form of tables or heat maps, but after a few attempts, we came to believe that these representations would not necessarily ease the understanding of the data. We agree with the Referee that 14 figures are many and that not all figures are absolutely necessary. Therefore, we have reviewed our figures again and came to the conclusion that Figures 5, 9, 11, 12 and 13 in the initial manuscript are less important for the mechanistic understanding of our findings. In other Journals, such figures would typically be placed in an online supplementary file, but this is not the policy of the Journal of Physiology. An option would be to write "(not shown)" for the corresponding results, but this would deprive the interested readers from information they may find beneficial. Therefore, we have incorporated an Appendix at the end of the article into which less important figures were moved and refer to these figures as Appendix Figures A1, A2, A3, A4 and A5. The other figures were renumbered accordingly. In this way, the main results section with its 9 figures forms a concise body of the most important results, leaving the choice to the reader to go through the appendix figures or not. We hope that this approach will be a good compromise between the completeness of the results, the ease of reading and the policy of the Journal of Physiology.

Dear Dr Kucera,

Re: JP-RP-2021-282105R1 "Localization of Na⁺ channel clusters in narrowed perinexi of gap junctions enhances cardiac impulse transmission via ephaptic coupling: a model study" by Ena Ivanovic and Jan P Kucera

I am pleased to tell you that your paper has been accepted for publication in The Journal of Physiology.

NEW POLICY: In order to improve the transparency of its peer review process The Journal of Physiology publishes online as supporting information the peer review history of all articles accepted for publication. Readers will have access to decision letters, including all Editors' comments and referee reports, for each version of the manuscript and any author responses to peer review comments. Referees can decide whether or not they wish to be named on the peer review history document.

Are you on Twitter? Once your paper is online, why not share your achievement with your followers. Please tag The Journal (@jphysiol) in any tweets and we will share your accepted paper with our 23,000+ followers!

The last Word version of the paper submitted will be used by the Production Editors to prepare your proof. When this is ready you will receive an email containing a link to Wiley's Online Proofing System. The proof should be checked and corrected as quickly as possible.

Authors should note that it is too late at this point to offer corrections prior to proofing. The accepted version will be published online, ahead of the copy edited and typeset version being made available. Major corrections at proof stage, such as changes to figures, will be referred to the Reviewing Editor for approval before they can be incorporated. Only minor changes, such as to style and consistency, should be made a proof stage. Changes that need to be made after proof stage will usually require a formal correction notice.

All queries at proof stage should be sent to TJP@wiley.com

Yours sincerely,

Bjorn Knollmann
Senior Editor
The Journal of Physiology

P.S. - You can help your research get the attention it deserves! Check out Wiley's free Promotion Guide for best-practice recommendations for promoting your work at www.wileyauthors.com/eo/guide. And learn more about Wiley Editing Services which offers professional video, design, and writing services to create shareable video abstracts, infographics, conference posters, lay summaries, and research news stories for your research at www.wileyauthors.com/eo/promotion.

* IMPORTANT NOTICE ABOUT OPEN ACCESS *

Information about Open Access policies can be found here
<https://physoc.onlinelibrary.wiley.com/hub/access-policies>

To assist authors whose funding agencies mandate public access to published research findings sooner than 12 months after publication The Journal of Physiology allows authors to pay an open access (OA) fee to have their papers made freely available immediately on publication.

You will receive an email from Wiley with details on how to register or log-in to Wiley Authors Services where you will be able to place an OnlineOpen order.

You can check if your funder or institution has a Wiley Open Access Account here
<https://authorservices.wiley.com/author-resources/Journal-Authors/licensing-and-open-access/open-access/author-compliance-tool.html>

Your article will be made Open Access upon publication, or as soon as payment is received.

If you wish to put your paper on an OA website such as PMC or UKPMC or your institutional repository within 12 months of publication you must pay the open access fee, which covers the cost of publication.

OnlineOpen articles are deposited in PubMed Central (PMC) and PMC mirror sites. Authors of OnlineOpen articles are permitted to post the final, published PDF of their article on a website, institutional repository, or other free public server, immediately on publication.

Note to NIH-funded authors: The Journal of Physiology is published on PMC 12 months after publication, NIH-funded authors DO NOT NEED to pay to publish and DO NOT NEED to post their accepted papers on PMC.

EDITOR COMMENTS

Reviewing Editor:

Congratulations on a well done and insightful study!

Senior Editor:

Congratulation for an exceptional modeling study that will be of major impact in the field.

REFEREE COMMENTS

Referee #1:

The authors have nicely addressed all of my concerns and comments.

Referee #2:

The authors have developed a responsive revision. It is a superb study and the authors should be commended.

END OF COMMENTS